# Ubiquitination of VE-cadherin regulates inflammation-induced vascular permeability in vivo

Markus Wilkens [iD], Leonie Holtermann [iD], Ann-Kathrin Stahl [iD], Rebekka I Stegmeyer [iD], Astrid F Nottebaum [iD] & Dietmar Vestweber [iD] ✉

## Abstract

VE-cadherin is a major component of the cell adhesion machinery which provides integrity and plasticity of the barrier function of endothelial junctions. Here, we analyze whether ubiquitination of VE-cadherin is involved in the regulation of the endothelial barrier in inflammation in vivo. We show that histamine and thrombin stimulate ubiquitination of VE-cadherin in HUVEC, which is completely blocked if the two lysine residues K626 and K633 are replaced by arginine. Similarly, these mutations block histamine-induced endocytosis of VE-cadherin. We describe two knock-in mouse lines with endogenous VE-cadherin being replaced by either a VE-cadherin K626/633R or a VE-cadherin KallR mutant, where all seven lysine residues are mutated. Mutant mice are viable, healthy and fertile with normal expression levels of junctional VE-cadherin. Histamine- or LPS-induced vascular permeability in the skin or lung of both of these mutant mice are clearly and similarly reduced in comparison to WT mice. Additionally, we detect a role of K626/633 for lysosomal targeting. Collectively, our findings identify ubiquitination of VE-cadherin as important for the induction of vascular permeability in the inflamed skin and lung.

**Keywords** VE-cadherin; Ubiquitination; Endothelial Junctions; Vascular Permeability
**Subject Categories** Cell Adhesion, Polarity & Cytoskeleton; Immunology; Post-translational Modifications & Proteolysis

## Introduction

Endothelial cells form a barrier between blood and interstitium. Common to many diseases, inflammatory processes and stimuli reduce the endothelial barrier function and increase plasma leaks by compromising endothelial junction integrity (Mehta and Malik, 2006; Claesson-Welsh et al, 2021). Understanding the mechanisms behind this requires more insights into the regulation of junctional cell adhesion.

VE-cadherin is a cell adhesion molecule that represents a major constituent of endothelial adherens junctions (Dejana and Vestweber, 2013). Evidence for the importance of VE-cadherin is based on the fact that interference with VE-cadherin by adhesion blocking antibodies or by conditional genetic inactivation is sufficient to increase vascular permeability in vivo in some organs (Gotsch et al, 1997; Corada et al, 1999; Frye et al, 2015). In addition, it was shown that enhancing the adhesive function of VE-cadherin in mutant knock-in mice leads to stabilized endothelial junctions and strongly increased barrier function (Schulte et al, 2011; Broermann et al, 2011).

The function of VE-cadherin is based on its linkage to the actin cytoskeleton, which is mediated by the catenins (Huber et al, 1996; Lampugnani et al, 1995; Huber et al, 2001; Vestweber, 2008) and further modulated by additional adaptor proteins, such as vinculin and EPLIN (Huveneers et al, 2012; Chervin-Pétinot et al, 2012; Taha et al, 2019). It is conceivable that this linkage or other aspects of the cell adhesive function of VE-cadherin may be addressed by mechanisms that interfere with endothelial junction integrity. Little is known, though, about such mechanisms in the context of inflammation, except for a potential role for tyrosine phosphorylation of β-catenin and plakoglobin (Chen et al, 2012; Nottebaum et al, 2008).

Phosphorylation of various amino acids of VE-cadherin was also reported to play a role in the regulation of endothelial junctions. Ser665 was shown to be involved in VEGF-induced endocytosis of VE-cadherin by a β-arrestin dependent mechanism (Gavard and Gutkind, 2006). Several tyrosine residues were reported to be involved in the regulation of permeability or leukocyte transmigration across cultured endothelial cell monolayers (Monaghan-Benson and Burridge, 2009; Wallez et al, 2007; Allingham et al, 2007; Turowski et al, 2008). Polyclonal antibodies against pY685 and pY658 of VE-cadherin were reported to stain VE-cadherin in a shear force dependent way specifically in venules but not in arterioles and this staining was reduced upon bradykinin treatment in venules at sites that correlated with vascular leaks (Orsenigo et al, 2012). A more direct analysis of the role of pY residues of VE-cadherin was based on a study with point mutated knock-in mice where VE-cadherin was replaced by a Y685F or a Y731F VE-cadherin mutant (Wessel et al, 2014). This study revealed that phosphorylation of Y685 selectively regulated the induction of vascular permeability whereas Y731 was selectively relevant only for the leukocyte diapedesis process. For the permeability inducing mechanism, it was shown that downstream of the phosphorylation of Y685, bradykinin induced ubiquitination of VE-cadherin, which

Max Planck Institute for Molecular Biomedicine, D-48149 Muenster, Germany. ✉E-mail: vestweb@mpi-muenster.mpg.de

was proposed to be involved in its endocytosis (Orsenigo et al, 2012). However, direct evidence for a role of ubiquitination of VE-cadherin in endocytosis upon stimulation with an inflammatory mediator was not presented.

Ubiquitination of membrane proteins controls their uptake into cells via endocytosis, as well as sorting and trafficking within cells (Foot et al, 2017). Ubiquitination occurs primarily on lysine residues and can generally influence protein targeting and cell adhesion by controlling cell surface expression of cell adhesion molecules (Majolée et al, 2019). Despite the importance of the ubiquitination system for the regulation of endocytosis, constitutive endocytosis can also be regulated independently based on other mechanisms and protein motifs, as was shown for VE-cadherin (Nanes et al, 2012).

Various E3 ubiquitin ligases have been reported to target VE-cadherin and affect endothelial junction integrity, such as the ligase K5 of Sarcoma virus (Nanes et al, 2017), March3 (Seo et al, 2023) as well as CHFR (Tiruppathi et al, 2013). Endothelial specific gene inactivation of CHFR was shown to block endotoxin-induced permeability in the mouse lung as well as downregulation of VE-cadherin, although a direct role of VE-cadherin ubiquitination in vivo in this process was not shown (Tiruppathi et al, 2023).

In the present study, we sought to determine whether ubiquitination of VE-cadherin is indeed required for the induction of VE-cadherin endocytosis by inflammatory mediators and whether it is needed for the induction of vascular permeability in vivo. To this end, we identified the essential lysine residues within the cytoplasmic domain of VE-cadherin which are responsible for the induction of ubiquitination and endocytosis of VE-cadherin by inflammatory mediators. Based on this, we generated point mutated knock-in mice by replacing VE-cadherin by the appropriate K/R point mutants of VE-cadherin, carrying arginine instead of certain lysine residues. Analyzing these mice in two vascular leak models, we could show that ubiquitination of VE-cadherin is in vivo required for histamine- and LPS-induced vascular permeability.

# Results

## Inflammation-induced ubiquitination of VE-cadherin targets K626 and K633

It has been shown that the phosphorylation of Y685 of VE-cadherin is involved in the induction of vascular permeability in vitro (Orsenigo et al, 2012) and in vivo (Wessel et al, 2014). In addition, it was shown that the Y685F point mutation of VE-cadherin interfered with bradykinin-induced ubiquitination of VE-cadherin (Orsenigo et al, 2012).

To analyze the relevance of VE-cadherin ubiquitination for the regulation of endothelial junctions in inflammation, we sought to identify which of the seven lysine residues (Fig. 1A) are relevant. To this end we generated point mutated constructs of VE-cadherin with the following lysine (K) residues replaced by arginine (R): K626R, K633R, K672R, K768R, K690R and the double mutants K626/633R and K712/713R and a mutant with all 7 lysine residues replaced by arginine residues (KallR). Each of these mutants as well as WT VE-cadherin was fused to EGFP at its C-terminus. To test the effect of these mutations on histamine-induced ubiquitination

of VE-cadherin in endothelial cells, we first depleted endogenous expression of VE-cadherin in HUVEC by siRNA, targeting the 5' upstream region (Fig. EV1A), followed 24 h later by adenovirus-based expression of one of the mutated VE-cadherin constructs. As shown by immunoblotting of anti-EGFP-immunoprecipitated VE-cadherin constructs (Fig. 1B,C), treatment of HUVEC with histamine (100 μM) for 10 min increased ubiquitination of WT VE-cadherin and of most of the mutated constructs except for the K626/633R and the KallR constructs. Mutation of the single K633 amino acid had a partial inhibitory effect on ubiquitination.

To test, whether other inflammatory stimuli would target the same lysine residues, we repeated the experiments with thrombin (VE-cadherin knock-down shown in Fig. EV1B). As shown in Fig. 1D,E, the results we had seen for histamine were confirmed for thrombin comparing WT VE-cadherin with the KallR and the K626/633R mutants. Again, the analyzed lysine point mutations blocked ubiquitination in each case.

Next, we tested the functional relevance of ubiquitination at lysine residues K626 and K633 for permeability regulation. To this end, we depleted endogenous VE-cadherin in HUVEC via siRNA (Fig. EV1C), seeded cells on transwell filters and transduced these cells with either WT VE-cadherin-EGFP or K626/633R VE-cadherin-EGFP followed by thrombin treatment. As shown in Fig. 1F, the significant increase of thrombin-induced permeability for 250 kDa FITC-dextran across WT VE-cadherin expressing cells was blocked by the K626/633R mutation (Fig. 1F). We conclude that the K626 and K633 residues are essential target sites for ubiquitination by the tested range of inflammatory mediators and ubiquitination at these two lysine residues is needed for efficient permeability induction in HUVEC monolayers.

## VE-cadherin endocytosis induced by inflammatory mediators requires K626 and K633

Next, we tested whether the identified lysine residues of VE-cadherin are indeed relevant for histamine-induced endocytosis of VE-cadherin. To this end, we silenced endogenous VE-cadherin in HUVEC by siRNA (Fig. EV2A) and transduced the cells 24 h later with WT VE-cadherin-EGFP or the corresponding Y685F and K626/633R mutant forms of VE-cadherin-EGFP. To distinguish histamine-induced endocytosis of VE-cadherin from intracellular VE-cadherin molecules that might be on their way to the cell surface, we incubated cells with a non-adhesion blocking antibody against VE-cadherin (55-7H1) prior to the incubation with histamine, then added histamine for 60 min, followed by an acid wash with 100 mM glycine-HCl, pH 2.5, to remove cell-surface-bound antibodies. Cells were then fixed and processed for immunofluorescence staining. Endocytosis was subsequently analyzed by automated counting of the number of 55-7H1-labeled intracellular vesicles per image and normalization to the mean EGFP fluorescence intensity in Fiji-ImageJ. We found a significant increase of histamine-induced endocytosis of WT VE-cadherin, whereas this effect was blocked by the Y685F and the K626/633R mutations (Fig. 2A). Quantification of the results is shown in Fig. 2B. Additionally, we also investigated the effect of the two mutations on VE-cadherin internalization by a biochemical approach, where again HUVEC were depleted of endogenous VE-cadherin by siRNA transfection (Fig. EV2B) followed by adenoviral re-expression of WT VE-cadherin-EGFP or the corresponding

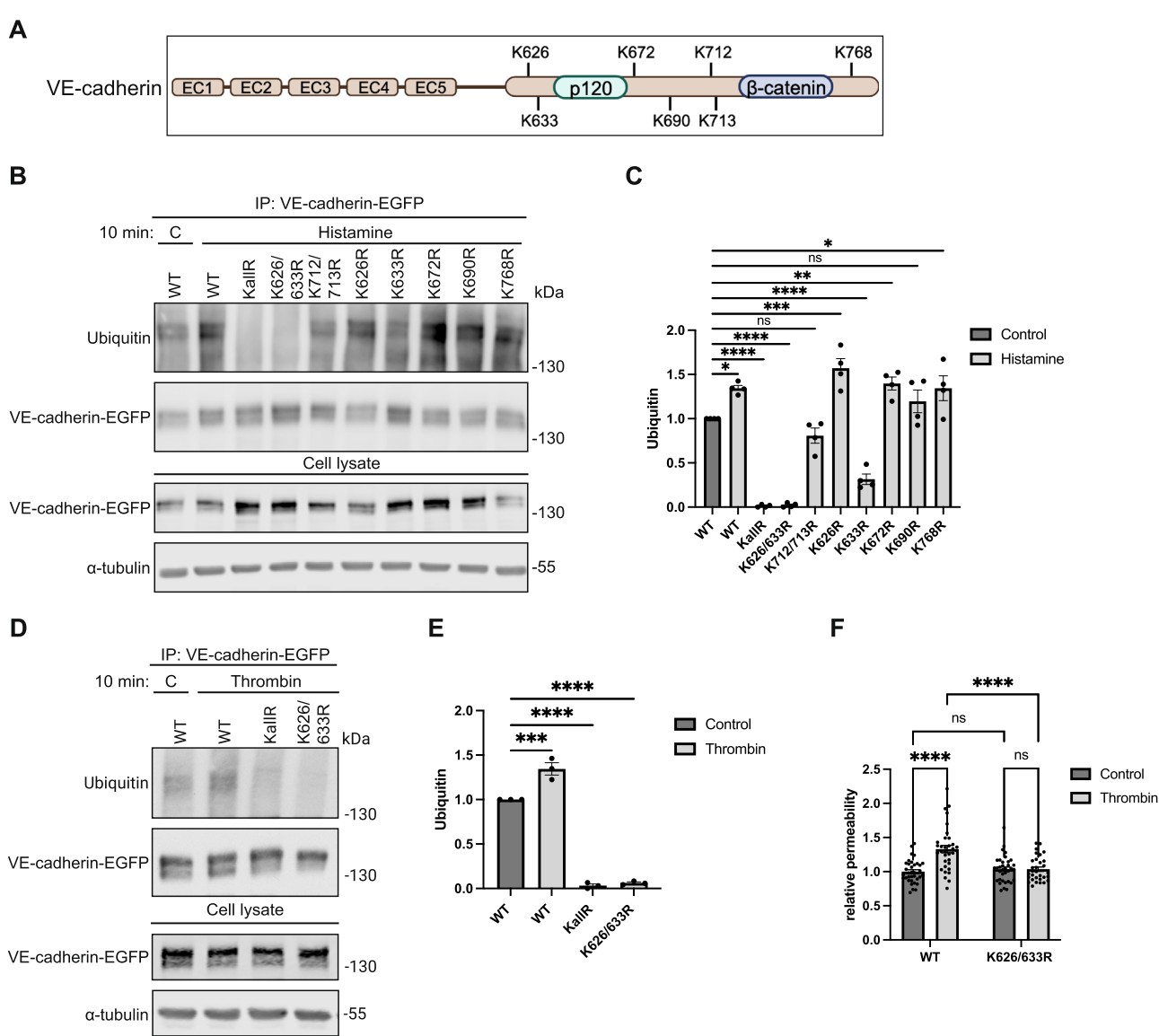

**Figure 1. Stimulus-dependent ubiquitination of VE-cadherin lysine residues K626 and K633.**

(A) Scheme of VE-cadherin domain structure indicating the position of all lysine residues within the cytoplasmic tail and the binding areas for p120 and β-catenin. Created with BioRender.com. (B) Upon silencing of endogenous VE-cadherin for 24 h, HUVEC were transduced with VE-cadherin-EGFP constructs carrying various lysine to arginine (K/R) mutations (as indicated), followed 48 h later by 10 min stimulation with histamine. Subsequently, immunoprecipitates (IP) of VE-cadherin-EGFP or cell lysates were immunoblotted for ubiquitin, EGFP or tubulin, as indicated on the left. Molecular weights are indicated in kilodaltons (kDa). (C) Quantification of immunoblot signals in (B), presented as relative ubiquitin levels normalized to VE-cadherin-EGFP. n = 4 biological replicates. (D) Set up as in (B), and treatment of HUVEC with thrombin for 10 min followed by immunoprecipitation and immunoblotting for Ubiquitin, VE-cadherin-EGFP and α-tubulin. (E) Quantification of immunoblot in (D), presented as relative ubiquitin levels normalized to VE-cadherin-EGFP. n = 3 biological replicates. (F) Paracellular permeability for 250 kDa FITC-dextran was determined for adenoviral transduced HUVEC monolayers. Upon silencing of endogenous VE-cadherin for 24 h, HUVEC were seeded on transwell filters, transduced with VE-cadherin-EGFP WT or K626/K633R mutant and grown to confluency for 48 h. Then paracellular permeability was measured after 30 min thrombin treatment. n = 4 biological replicates. Data information: In (C, E, F), data are presented as mean ± SEM. *P < 0.05; **P < 0.01; ***P < 0.001; ****P < 0.0001; ns, not significant. One-way ANOVA with Holm-Sidak multiple comparison test in (C, E) or two-way ANOVA with Tukey's multiple comparison test in (F). P = 0.0208 (WT C vs. WT His), P = 0.00000001 (WT C vs. KallR His), P = 0.00000001 (WT C vs. K626/633R His), P = 0.1832 (WT C vs. K712/713 R His), P = 0.0001 (WT C vs. K626R His), P = 0.00001 (WT C vs. K633R His), P = 0.0077 (WT C vs. K672R His), P = 0.1832 (WT C vs. K690R His) and P = 0.0208 (WT C vs. K768R His) in (C). P = 0.0002 (WT C vs. WT T), P = 0.0000002 (WT C vs. KallR T), P = 0.0000002 (WT C vs. K626/633R T) in (E). P = 0.00000007 (WT C vs. WT T) and P = 0.000002 (WT T vs. K626/633R T) in (F). Source data are available online for this figure.

Y685F or K626/633R mutants. Subsequently, the cells were incubated with a cell-impermeable biotinylation reagent for 30 min, followed by treatment with histamine for 60 min. After cleavage of surface-bound biotin, the cells were lysed and internalized proteins were precipitated with Neutravidin agarose beads. Western blot analysis of the levels of internalized VE-cadherin revealed that while histamine stimulation resulted in a significant increase in the amount of endocytosed VE-cadherin in cells expressing WT VE-cadherin, no such effect was evident for cells expressing either of the mutant constructs (Fig. 2C,D). These

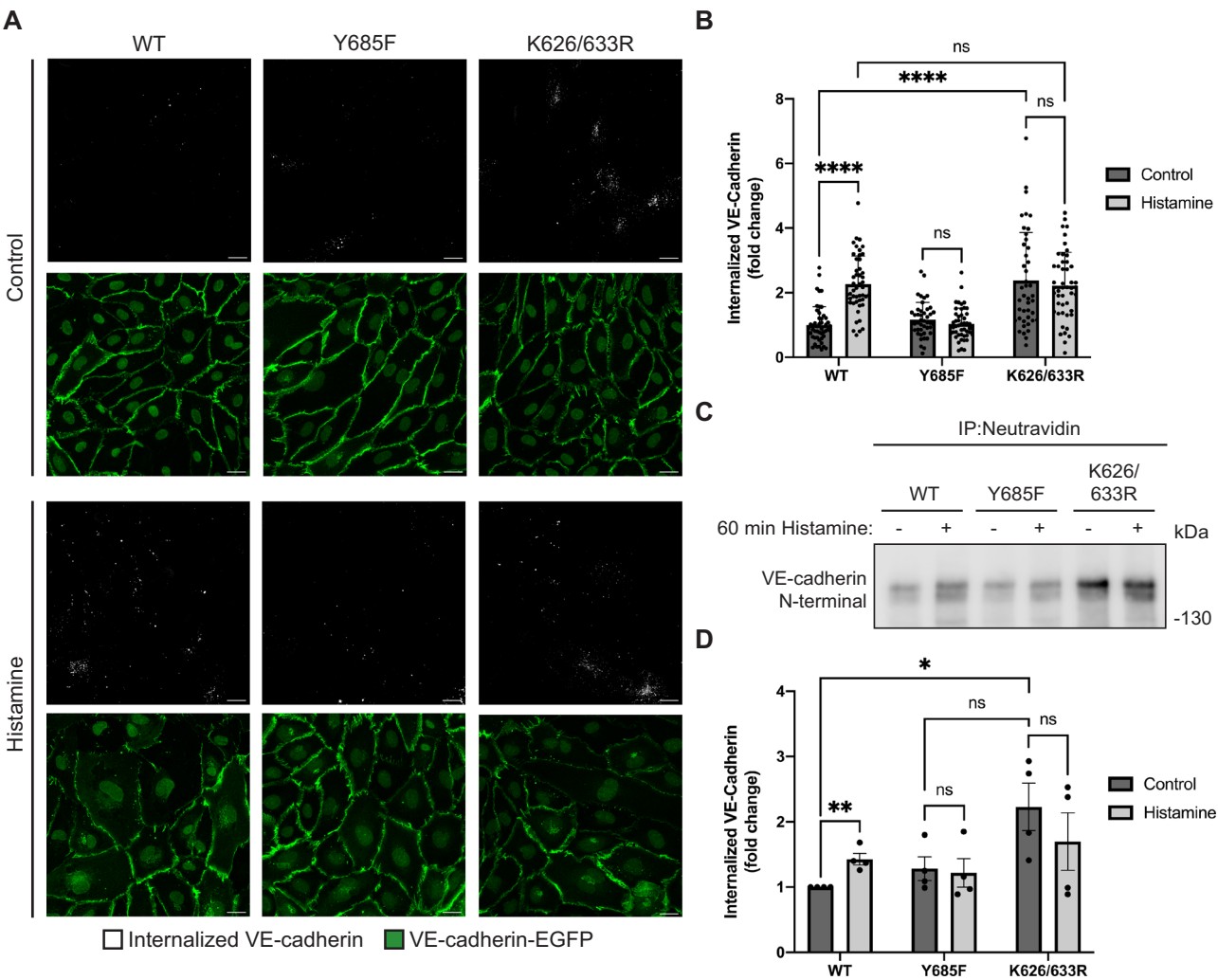

**Figure 2. Histamine-induced endocytosis of VE-cadherin depends on K626 and K633.**

(A) HUVECs were depleted of endogenous VE-cadherin by siRNA transfection, followed by adenoviral re-expression of EGFP-tagged VE-cadherin WT, Y685F or K626/633R constructs 24 h later. HUVECs were then incubated with an antibody directed against the N-terminus of VE-cadherin (55-7H1) and stimulated with histamine for 60 min, followed by an acid wash, fixation and immunofluorescent staining. (B) Histamine-induced VE-cadherin internalization (as analyzed in (A)) was quantified from the number of intracellular vesicles positive for 55-7H1 and normalized to mean VE-cadherin-EGFP intensity. $n = 3$ biological replicates, using 15–20 images per construct per experiment. (C) HUVECs were depleted of endogenous VE-cadherin by siRNA transfection, followed by adenoviral re-expression of EGFP-tagged VE-cadherin WT, Y685F or K626/633R constructs 24 h later. 100 mm dishes of subconfluent HUVECs were incubated with a cell-impermeable biotinylation reagent followed by treatment with histamine or vehicle for 60 min. Cell surface-bound biotin was cleaved and cells were lysed. Clarified lysates were subjected to immunoprecipitation with neutravidin agarose beads and analyzed by Western Blot with an anti-VE-cadherin antibody directed against the N-terminus of the protein (AF938). (D) Quantification of VE-cadherin internalization (as analyzed in (C)) normalized to WT control. $n = 4$ biological replicates. Data information: In (B, D), data are presented as mean ± SEM. *$P < 0.05$; **$P < 0.01$; ****$P < 0.0001$; ns, not significant. Two-way ANOVA with Tukey's multiple comparisons test in (B) and unpaired two-tailed $t$ test in (D). $P = 0.0000000003$ (WT control vs. WT histamine) and $P = 0.00000000002$ (WT control vs. K626/633R control) in (B). $P = 0.009135$ (WT control vs. WT histamine) and $P = 0.029487$ (WT control vs. K626/633R control) in (D). Scale bars: 20 μm in (A). Source data are available online for this figure.

results indicate that histamine-induced endocytosis of VE-cadherin is indeed impaired by mutating Y685 and requires the ubiquitination of lysine residues 626 and 633.

Although mutating K626 and K633 blocked histamine-induced endocytosis of VE-cadherin, we detected a higher level of intracellular VE-cadherin-K626/633R than of WT VE-cadherin in cells that were not histamine-stimulated. As will be shown below, this is most likely due to impaired lysosomal targeting and degradation of VE-cadherin-K626/633R which was constitutively internalized during the course of the assay.

## Generation of K626/633R and KallR point mutated VE-cadherin knock-in mice

To elucidate the role of VE-cadherin ubiquitination for inflammation-induced vascular permeability in vivo and to identify the relevant lysine residues, we generated two different point mutated knock-in mouse lines by replacing endogenous VE-cadherin by either the VE-cadherin K626/633R double mutant or the KallR mutant (Fig. 3A). Homologous recombination was achieved by recombinase-mediated cassette exchange similarly as

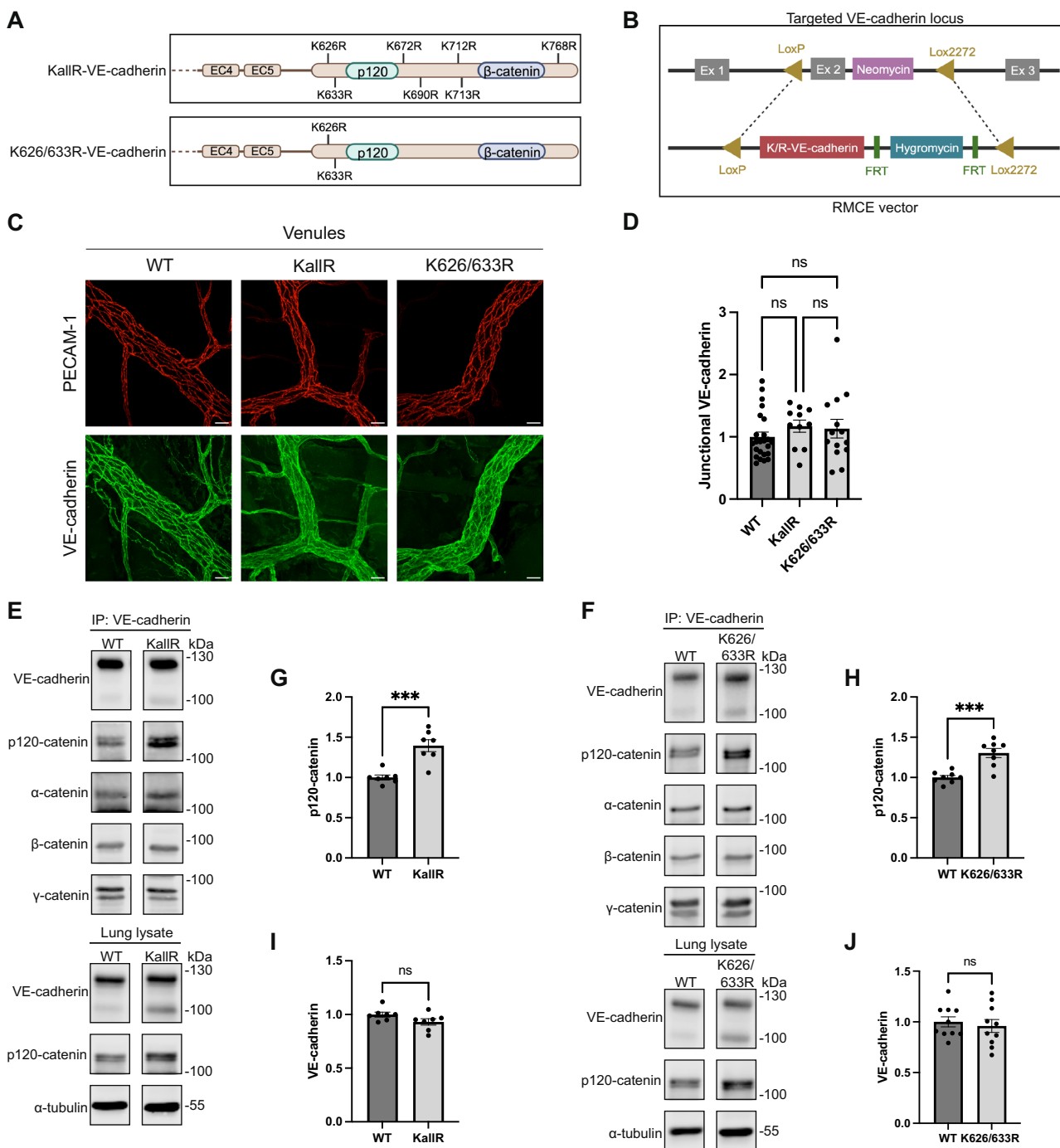

described for other VE-cadherin knock-in mouse lines (Schulte et al, 2011; Broermann et al, 2011; Wessel et al, 2014) (Fig. 3B). The point mutated VE-cadherin K626/633R and VE-cadherin KallR mice resulting from this approach were viable, fertile, and healthy and developed normally. The mutant allele was inherited with the expected Mendelian frequency (Tables 1–3). Expression levels and junction recruitment of both mutant forms of VE-cadherin were similar as for the knock-in of WT VE-cadherin, as was tested by whole mount antibody staining of the cremaster muscle (Fig. 3C). Quantification of junctional VE-cadherin was performed as

described in Material and Methods and revealed no difference in expression levels (Fig. 3D). The WT knock-in mouse line has been described before and expresses similar levels of VE-cadherin as WT C57Bl/6 mice (Schulte et al, 2011). Expression levels of VE-cadherin in the different mouse lines were also determined for the lung by comparing VE-cadherin immunoblot signals of lung lysates and of VE-cadherin immunoprecipitates from lung lysates (Fig. 3E,F). Again, quantification revealed no difference in expression levels of WT VE-cadherin or the two mutant forms of VE-cadherin (Fig. 3I,J).

**Figure 3. Design and characterization of knock-in mice expressing VE-cadherin KallR and K626/633R.**

(A) Design of the two VE-cadherin mutants with the respective lysine residues in the cytoplasmic tail mutated to arginine residues. Created with BioRender.com. (B) Strategy for recombinase-mediated cassette exchange (RMCE) to target VE-cadherin exon 2, which contains the ATG start codon, within the VE-cadherin gene locus. Previously, ES cells were generated with two incompatible lox sites, loxP and lox2272, flanking exon 2. This exon can be replaced with the help of the recombinase Cre by a LoxP/Lox2272 flanked cassette that contains a VE-cadherin cDNA bearing the desired mutations as well as a hygromycin gene (for selection) flanked by FRT sites. Created with BioRender.com. (C) Whole mount staining of venules in the cremaster muscle from homozygous WT, KallR and K626/633R mice was performed using antibodies against VEcadherin and PECAM-1, as indicated. Images are representative of three mice per group. (D) Quantification of junctional VE-cadherin (as analyzed in (C)), presented as relative junctional VE-cadherin level normalized to vessel volume. Data are representative of at least ten vessels per genotype. (E, F) Immunoblot analysis of either anti-VEcadherin immunoprecipitates (C5 antibody) from lung lysates or directly of lung lysates from homozygous WT or VE-cadherin KallR mice (E) or from VE-cadherin-K626/633R mice (F), using antibodies against VE-cadherin (AF1002 antibody), p120-catenin, α-catenin, β-catenin, γ-catenin, and α-tubulin, as indicated. (G, H) Quantification of p120-catenin levels co-precipitated with VE-cadherin (from experiments in (E, F)), normalized to immunoprecipitated VE-cadherin. $n = 7$ mice/ group in (G) and $n = 8$ mice/group in (H). (I, J) Quantification of VE-cadherin in whole lung lysates (as detected in (E, F)), presented as relative VE-cadherin levels normalized to α-tubulin. $n = 7$ mice/group in (I) and $n = 10$ mice/group in (J). Data information: In (D, G, H, I, J), data are presented as mean ± SEM. ***$P < 0.001$; ns, not significant. One-way ANOVA with Holm–Sidak multiple comparison test in (D) or unpaired two-tailed *t* test in (G, H, I, J). $P = 0.0003$ (WT vs. KallR) in (G). $P = 0.0002$ (WT vs. K626/633R) in (H). Scale bars: 20 μm in (C). Source data are available online for this figure.

**Table 1. Genotype frequency in F1 and F2 generation of heterozygous K/R VE-cadherin matings.**

| Mouse line | Genotype | Total | Female | Male |
|---|---|---|---|---|
| KallR VE-cadherin #C9 (F1) | +/+ | 5 (18%) | 3 (60%) | 2 (40%) |
| | +/d | 14 (50%) | 1 (7%) | 13 (93%) |
| | d/d | 9 (32%) | 7 (78%) | 2 (22%) |
| K626/633R VE-cadherin #C9 (F2) | +/+ | 8 (24%) | 5 (64%) | 3 (38%) |
| | +/d | 16 (49%) | 6 (38%) | 10 (64%) |
| | d/d | 9 (27%) | 3 (33%) | 6 (67%) |

Wild type allele is designated with a + and RMCE knock-in allele with a d. Frequencies of the RMCE locus fall within the expected Mendelian segregation ratio.

**Table 2. Genotype frequency in F6 generation of heterozygous K/R VE-cadherin matings.**

| Mouse line | Genotype | Total | Female | Male |
|---|---|---|---|---|
| KallR VE-cadherin #C9 (F6) | +/+ | 11 (17%) | 2 (18%) | 9 (82%) |
| | +/d | 34 (53%) | 14 (41%) | 20 (59%) |
| | d/d | 19 (30%) | 10 (53%) | 9 (47%) |
| K626/633R VE-cadherin #C9 (F6) | +/+ | 17 (24%) | 10 (59%) | 7 (41%) |
| | +/d | 39 (55%) | 16 (41%) | 23 (59%) |
| | d/d | 15 (21%) | 6 (40%) | 9 (60%) |

Wild type allele is designated with a + and RMCE knock-in allele with a d. Frequencies of the RMCE locus fall within the expected Mendelian segregation ratio.

In addition to quantifying the VE-cadherin expression levels we also analyzed the levels of co-precipitated catenins. We found that similar levels of α-, β- and γ-catenin were co-precipitated with VE-cadherin in WT as well as in the two K/R mutant mouse lines (Fig. 3E,F). The only exception was the catenin p120 which was co-precipitated at higher levels from K626/633R and KallR mutant mice than from WT mice, arguing for a potential role of K626 and K633 ubiquitination in modulating p120 association to VE-cadherin in vivo (Fig. 3G,H).

**Table 3. Genotype frequency in F11 generation of heterozygous K/R VE-cadherin matings.**

| Mouse line | Genotype | Total | Female | Male |
|---|---|---|---|---|
| KallR VE-cadherin #C9 (F11) | +/+ | 13 (25%) | 4 (31%) | 9 (69%) |
| | +/d | 25 (47%) | 14 (56%) | 11 (44%) |
| | d/d | 15 (28%) | 9 (60%) | 6 (40%) |

Wild type allele is designated with a + and RMCE knock-in allele with a d. Frequencies of the RMCE locus fall within the expected Mendelian segregation ratio.

## Histamine- and LPS-induced vascular permeability are impaired in VE-cadherin K626/633R and KallR knock-in mice

To determine the role of ubiquitination of VE-cadherin for the induction of vascular permeability by inflammatory mediators in vivo, we performed Miles assays with both K/R mutant mouse lines and in WT VE-cadherin knock-in mice. We found that local intradermal injections of histamine in mice that had been i.v. injected with Evans blue induced strong leakiness in WT mice which was partially reduced in VE-cadherin K626/633R mice, as was found for mouse lines of mixed and of 98.4% C57Bl/6 background (2nd versus 6th backcross) (Fig. 4A,B). The partial inhibitory effect was not due to compensatory functions of other lysine residues in VE-cadherin, since also VE-cadherin KallR mice showed a similar inhibitory effect on permeability induction as K626/633R mice (Fig. 4C,D). Of note, histamine-induced vascular permeability was inhibited to a similar extent in VE-cadherin Y685F mice, when compared to WT mice (Nanes et al, 2012), which is in line with the concept that phosphorylation of Y685 and ubiquitination of K626 and K633 act in the same mechanistic pathway that contributes to the destabilization of endothelial junctions in vivo.

As a clinically significant model, LPS can trigger sustained inflammatory endothelial barrier dysfunction in the lung, whereas histamine-induced vascular permeability in the skin is rather transient. Thus, we decided to investigate if the K/R mutations of VE-cadherin do also affect vascular permeability in this clinically relevant model. For LPS challenge, mice were exposed to nebulized LPS for 40 min, and vascular permeability was determined by injecting Evans blue 5 h later for 30 min. As shown in Fig. 4E,F,

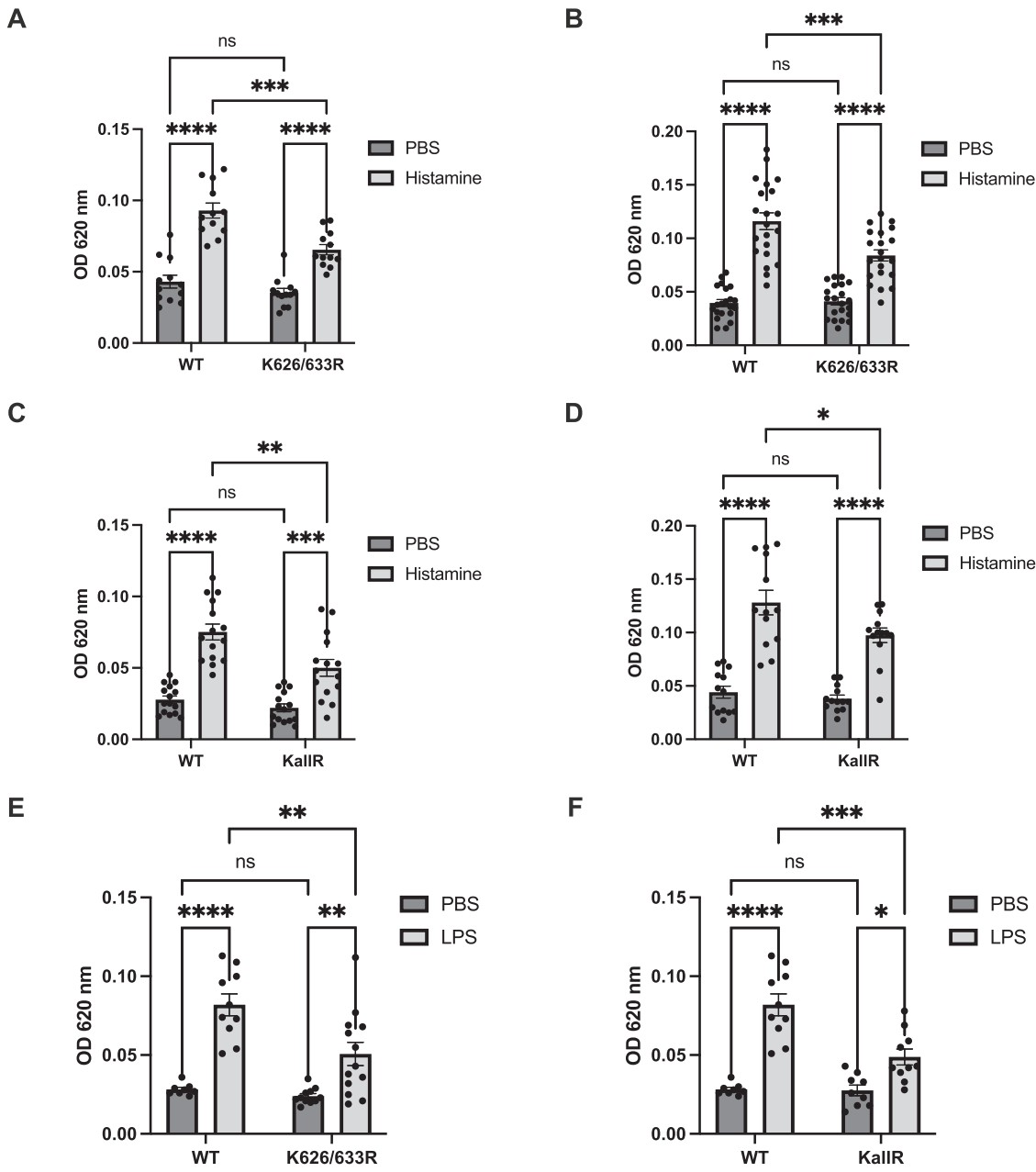

**Figure 4. Histamine- or LPS-induced vascular permeability is reduced in VE-cadherin K626/633R and KallR knock-in mice.**

(A–D) Vascular hyperpermeability in the skin of WT, KallR or K626/663R mice was analyzed with a Miles assay. 15 min after intravenous injection of Evans blue, PBS or histamine was given intradermally (as indicated) followed 30 min later by excision of skin areas and extraction and quantification of the leaked dye. $n = 12$ mice/group, homozygous WT littermates (F2) and K626/633R (F2) in (A), $n = 20$ mice/group, homozygous WT KI (C57Bl/6 background) and K626/633R (F6) in (B), $n = 15$ mice/group, homozygous WT KI (F1) and KallR (F1) in (C), $n = 13$ mice/group, homozygous WT KI (C57Bl/6 background) and KallR (F6 (5 mice) and F11 (8 mice), 13 mice/group in (D). (E, F) Vascular hyperpermeability in the lung of WT, KallR or K626/663R mice was analyzed upon LPS treatment. Mice were exposed to nebulized PBS or LPS for 40 min (as indicated). After 5 h, Evans blue was injected intravenously and 30 min later mice were sacrificed, the lung circulation was perfused, and the extracted dye from the lung tissue was quantified. $n = 8$ WT KI mice and 13 K626/633R mice in (E) and $n = 8$ WT KI mice and 10 KallRmice in (F). Data information: In (A–F), data are presented as mean ± SEM. $*P < 0.05$; $**P < 0.01$; $***P < 0.001$; $****P < 0.0001$; ns, not significant. Two-way ANOVA with Tukey's multiple comparison test in (A–F). $P = 0.0000000001$ (WT PBS vs. WT His), $P = 0.0002$ (WT His vs. K626/633R His) and $P = 0.00004$ (K626/633R PBS vs. K626/633R His) in (A). $P = 0.000000000001$ (WT PBS vs. WT His), $P = 0.0003$ (WT His vs. K626/633R His) and $P = 0.000001$ (K626/633R PBS vs. K626/633R His) in (B). $P = 0.000000003$ (WT PBS vs. WT His), $P = 0.0011$ (WT His vs. KallR His) and $P = 0.0002$ (KallR PBS vs. KallR His) in (C). $P = 0.000000001$ (WT PBS vs. WT His), $P = 0.026$ (WT His vs. KallR His) and $P = 0.000005$ (KallR PBS vs. KallR His) in (D). $P = 0.000003$ (WT PBS vs. WT LPS), $P = 0.0017$ (WT LPS vs. K626/633R LPS and $P = 0.0066$ (K626/633R PBS vs. K626/633R LPS) in (E). $P = 0.000003$ (WT PBS vs. WT LPS), $P = 0.0001$ (WT LPS vs. KallR LPS) and $P = 0.0219$ (KallR PBS vs. KallR LPS) in (F). OD 620: Optical density at 620 nm. Source data are available online for this figure.

vascular permeability in lung tissue of LPS treated VE-cadherin WT mice was strongly increased. This increase was clearly and significantly, although not completely, reduced in K626/633R and KallR mice.

Thus, vascular leakage induced either by histamine in the skin or LPS in the lung clearly depends on lysine ubiquitination of VE-cadherin, but can still be stimulated—although at significantly reduced levels—even in the absence of all lysine residues of VE-cadherin.

## Besides endocytosis, the K626/633R mutation of VE-cadherin impairs lysosomal clearing of intracellular VE-cadherin

When we compared the in vivo expression levels of VE-cadherin KallR and K626/633R with WT VE-cadherin by whole mount staining of venules in the cremaster, we localized the VE-cadherin mutant not only at junctions but also in intracellular vesicles, a location which we hardly detected for WT VE-cadherin (Fig. 5A,B). Interestingly, such vesicular staining was not detected in arterioles. Also in venules of diaphragm and cornea we could detect increased staining for VE-cadherin in intracellular vesicles of KallR and K626/633R mice (Fig. 6A,B). To allocate the VE-cadherin vesicles to intracellular compartments, we tested co-localization with markers for early endosomes (EE) or multivesicular bodies (MVB) by whole mount staining of cremaster venules as described in Material and Methods. As shown in Fig. 7A,B, we found increased co-localization of VE-cadherin vesicles and markers for EE and MVB in venules of KallR and K626/633R mice in comparison to venules of WT mice. To further analyze this phenomenon, we performed immunoblots for VE-cadherin with lung lysates of WT VE-cadherin, VE-cadherin-KallR and VE-cadherin-K626/633R mice. For both mutant mice we found an accumulation of a 100 kDa fragment of VE-cadherin in the lung lysates of these mice (Fig. 8A,C). This fragment was much more weakly seen in WT mice. Quantification of the blot signals revealed an increase of about threefold in comparison to WT mice (Fig. 8E,F). The 100 kDa fragment had been generated by cleaving off the C-terminus of VE-cadherin, since only antibodies against the N-terminus of VE-cadherin recognized the fragment, whereas an antibody against a C-terminal peptide of VE-cadherin did not (compare Fig. 8A and C with 8B and D).

The 100 kD fragment was reminiscent of a similar fragment of endocytosed VE-cadherin which we found to accumulate upon endothelial cell treatment with an inhibitor of lysosomal acidification. For this, we first silenced VE-cadherin by siRNA in HUVEC followed by transduction with either human WT-VE-cadherin-EGFP, VE-cadherin-K626/633R-EGFP or VE-cadherin-KallR-EGFP, then followed by treatment with the lysosomal inhibitor bafilomycin. As shown in Fig. 8G,H, we found that the 110 kDa fragment which had accumulated upon lysine mutation, increased even further in expression intensity due to the bafilomycin effect. We conclude that VE-cadherin which was constitutively endocytosed despite the lack of lysine residues is directed via the EE and MVB to the lysosome by a mechanism that is supported by the membrane proximal lysine residues 626 and 633. This interpretation also implies that constitutive endocytosis is still possible independent of lysine ubiquitination and that lysosomal targeting is supported, but not absolutely dependent on lysine ubiquitination.

## Discussion

In this study, we investigated the role of ubiquitination of VE-cadherin for the regulation of vascular permeability in inflammation. With K626 and K633 we detected two of the seven lysine residues within the cytoplasmic domain of VE-cadherin as essential sites for histamine and thrombin induced ubiquitination. Histamine-induced endocytosis of VE-cadherin and thrombin-induced permeability across HUVEC monolayers was also dependent on these two lysine residues. Replacing them by arginine in point-mutated VE-cadherin knock-in mice revealed that ubiquitination of these sites supports stimulation of vascular permeability by histamine in the skin and by LPS in the lung. Mutating all seven lysine residues in VE-cadherin in vivo had no stronger inhibitory effect on vascular permeability induction. We conclude that ubiquitination of K626 and K633 of VE-cadherin is needed for inflammation-induced vascular permeability in vivo.

Whereas our results reveal the importance of VE-cadherin ubiquitination and endocytosis for the induction of inflammatory leakage in vivo, these results show at the same time that additional mechanisms must exist which allow, at reduced efficiency, the induction of vascular permeability even in the absence of all lysine residues of VE-cadherin. Such additional mechanisms could be based on the modulation of VE-cadherin adhesive activity which could well act in WT mice synergistically with endocytosis-stimulating mechanisms. In addition, it is possible that other adhesion molecules at endothelial cell contacts are affected by permeability-inducing stimuli. Thus, our results allow to judge the significance of VE-cadherin ubiquitination and endocytosis for the regulation of endothelial junction integrity in vivo.

We have shown before that histamine and VEGF induce phosphorylation of Y685 of VE-cadherin in vitro and in vivo, and that VE-cadherin Y685F point-mutated knock-in mice are partly protected against histamine- and VEGF-induced vascular permeability in the skin (Wessel et al, 2014). Analyzing the downstream mechanism, it was found that bradykinin modulates Y685 phosphorylation of VE-cadherin and this tyrosine was required for bradykinin-induced endocytosis and ubiquitination of VE-cadherin (Orsenigo et al, 2012). However, the relevant lysine residues were not identified and it was not demonstrated that ubiquitination of VE-cadherin was indeed required for endocytosis induced by inflammatory mediators. The present study addresses these points and thereby demonstrates the relevance of VE-cadherin ubiquitination for histamine-induced endocytosis of VE-cadherin and for histamine-induced vascular permeability in vivo.

Regulation of the cell surface expression of cadherins has been analyzed in the context of embryonic development and tumor biology for various cadherins. E-cadherin endocytosis was found to be triggered by ubiquitination with the E3-ligase Hakai (Fujita et al, 2002). Kaposi's sarcoma associated herpesvirus was shown to induce VE-cadherin ubiquitination and endocytosis which was suggested to be relevant for vascular tumor development (Nanes et al, 2017). The human herpes virus 8 (HHV-8) encoded E3-ligase K5 was responsible for these effects. Interestingly, the lysine residues responsible for the virus derived K5 enzyme-driven endocytosis of VE-cadherin were the same two membrane proximal residues as we have identified here for histamine-induced VE-cadherin ubiquitination. K5 exhibits homology to human membrane-associated RING-CH type ubiquitin ligases

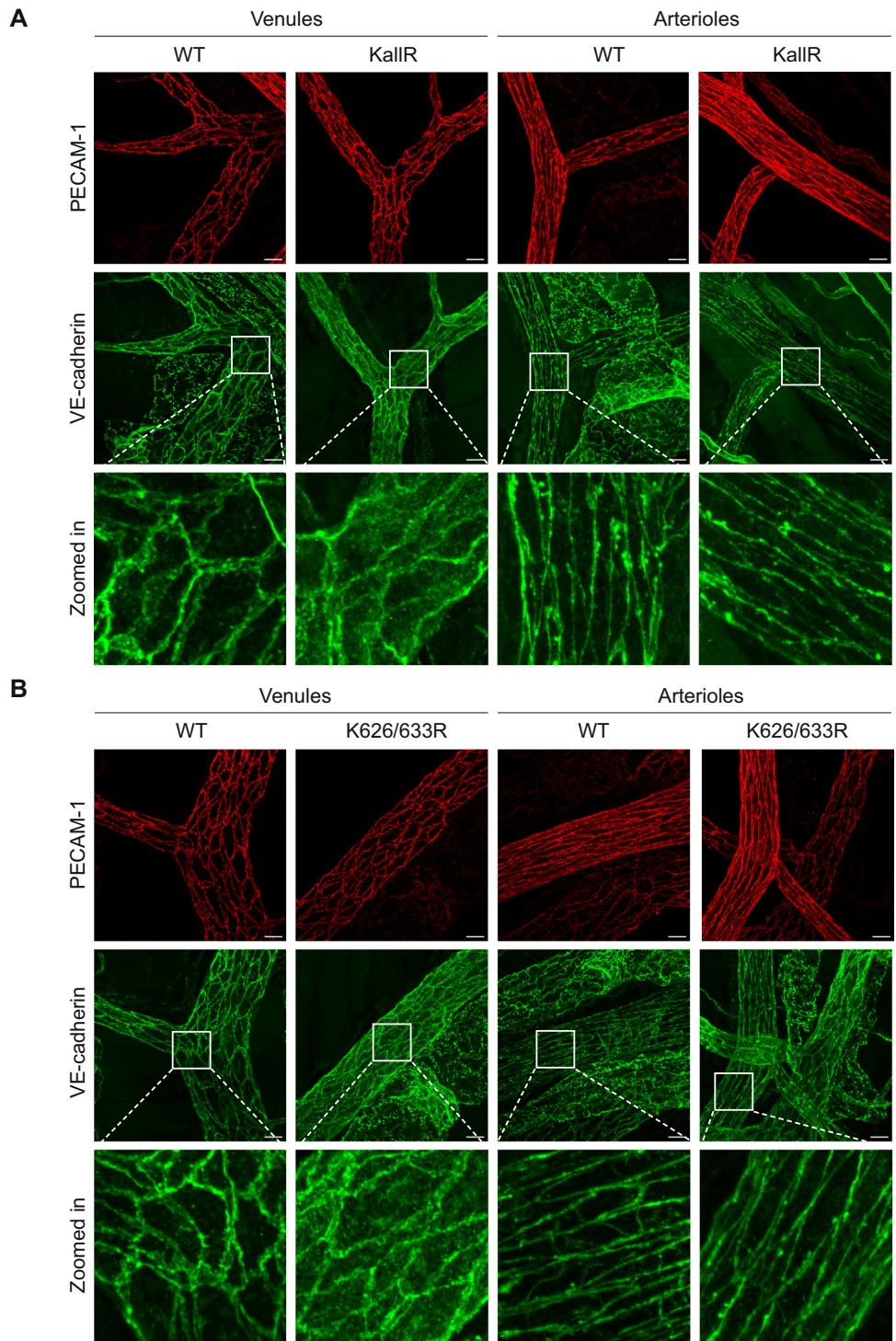

**Figure 5. Increased intracellular staining for VE-cadherin in cremaster venules of KallR and K626/633R knock-in mice.**

(A, B) Whole mount staining of venules and arterioles in cremaster muscles from homozygous WT, KallR (A) and K626/633R (B) mice was performed using antibodies against the N-terminus of VE-cadherin (AF1002) and PECAM-1. *n* = 3 mice/group. Data information: Scale bars: 20 μm in (A, B). Source data are available online for this figure.

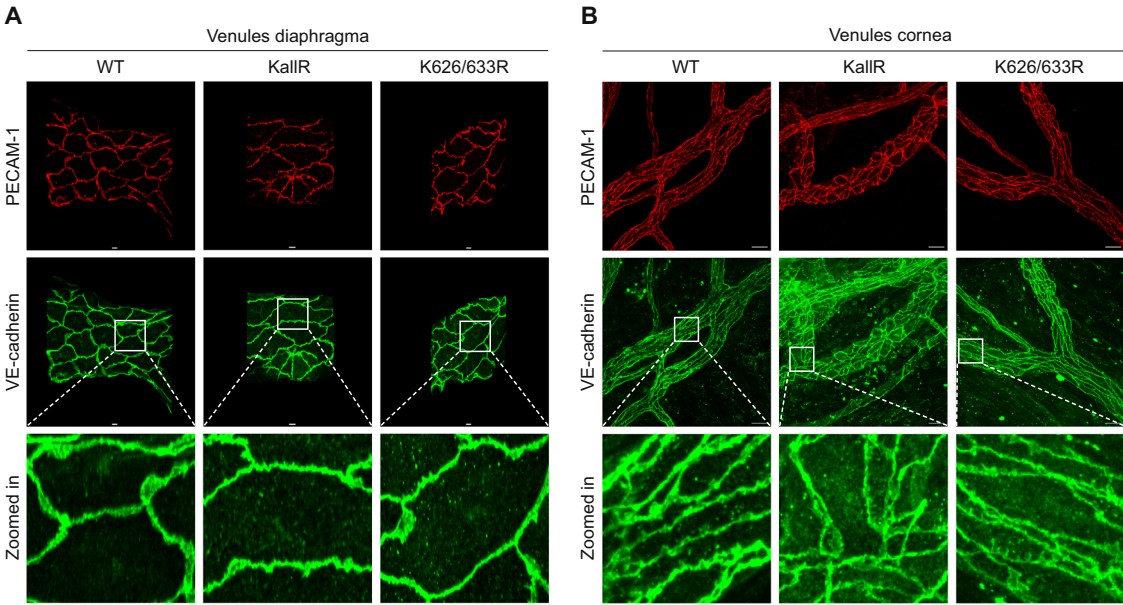

**Figure 6. Increased intracellular staining for VE-cadherin in diaphragm and cornea venules of KallR and K626/633R knock-in mice.**

(A, B) Whole mount staining of venules in diaphragm in (A) and cornea in (B) from homozygous WT, KallR and K626/633R mice was performed using antibodies against the N-terminus of VE-cadherin (AF1002) and PECAM-1. Data information: Scale bars: 10 µm in (A) and 20 µm in (B). Source data are available online for this figure.

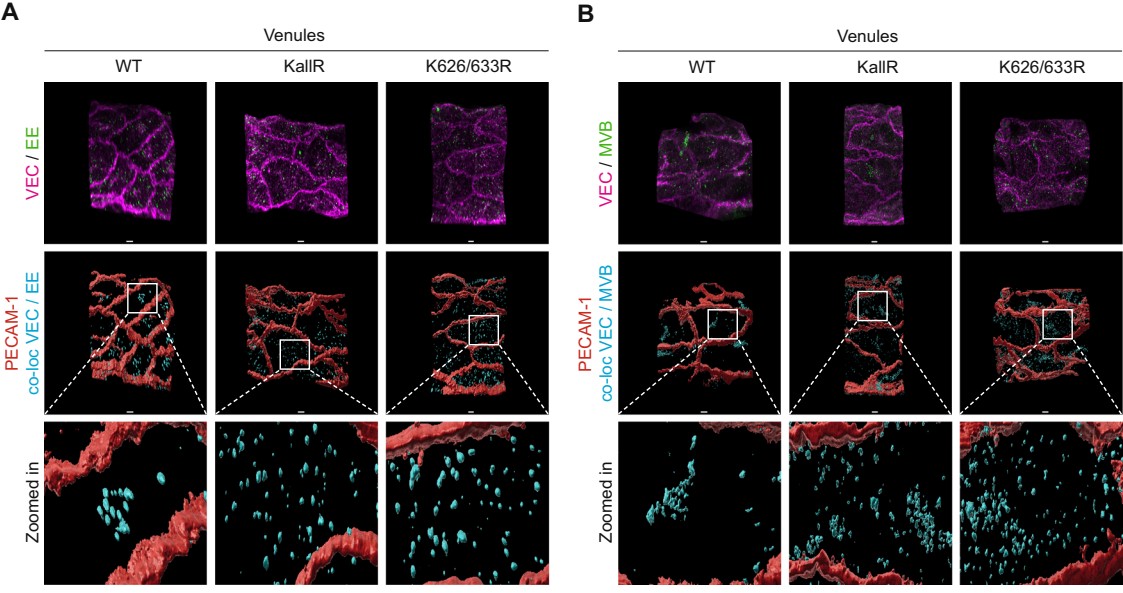

**Figure 7. Increased intracellular staining for VE-cadherin colocalizes with markers for early endosome and multivesicular body.**

(A, B) Whole mount staining of venules in cremaster muscles from homozygous WT, KallR and K626/633R mice was performed using antibodies against the N-terminus of VE-cadherin (VEC; AF1002), PECAM-1 and early endosome marker (EE) in (A) and multivesicular body marker (MVB) in (B). Calculated co-localization of VEC and EE (cyan) in (A) and of VEC and MVB (cyan) in (B). Data information: Scale bars: 5 µm in (A, B). Source data are available online for this figure.

(MARCH) (Ohmura-Hoshino et al, 2006). A member of this family, MARCH2, was recently reported to be expressed in endothelial cells and stimulate VE-cadherin endocytosis when transfected into HUVEC (Seo et al, 2023). Silencing of MARCH3 in HUVEC strengthened endothelial cell contacts, but mainly indirectly by increasing the expression of occludin, claudin5, and VE-cadherin, via modulating the stability of the FOXO1 transcription repressor (Leclair et al, 2016).

Recently, mice with an endothelial cell selective gene inactivation of the E3-ligase CHFR were reported to be completely resistant to LPS-induced vascular permeability in the lung (Tiruppathi et al,

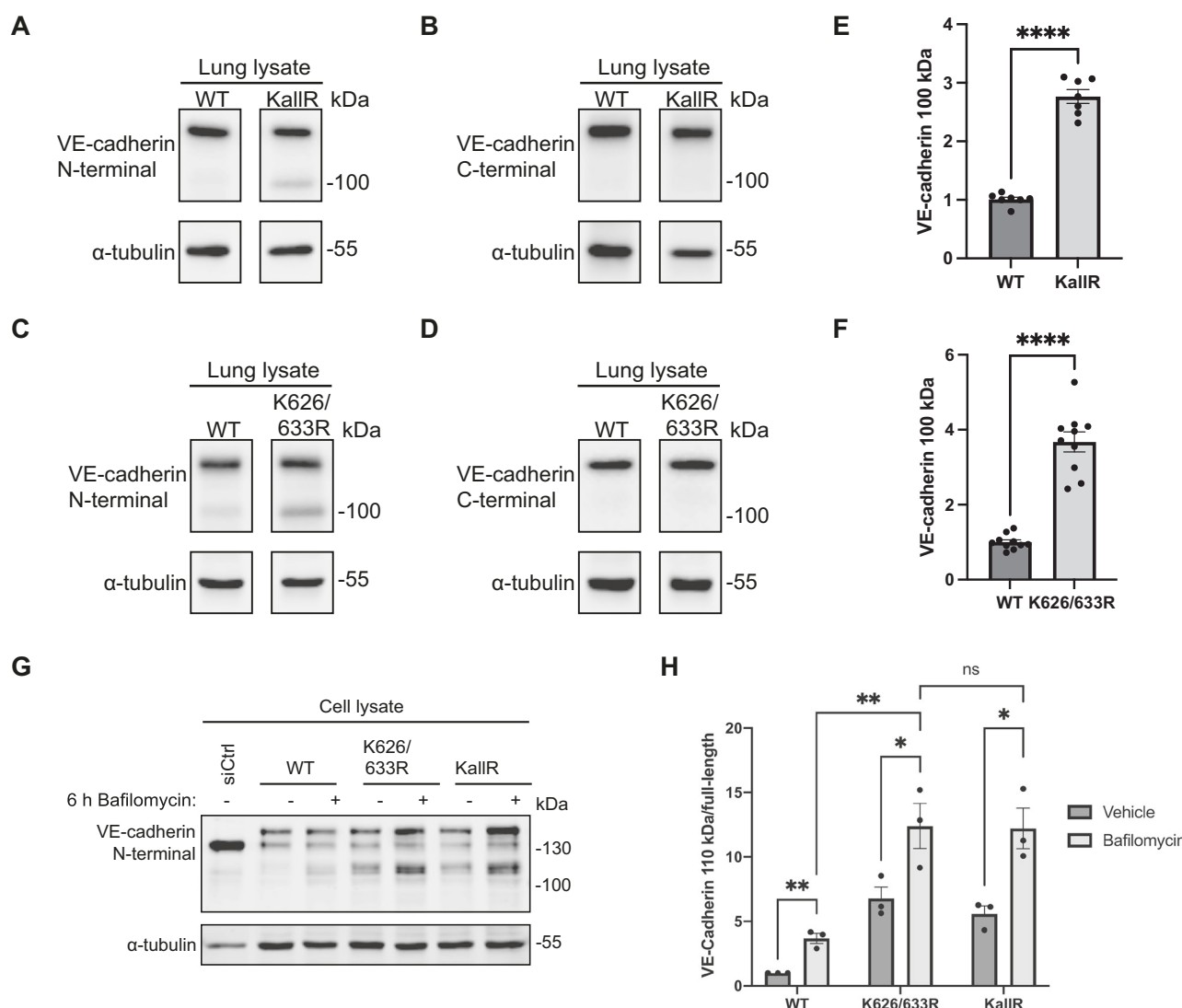

**Figure 8. Lysine ubiquitination supports lysosomal targeting of a 100 kDa N-terminal VE-cadherin fragment.**

(A–D) Immunoblot analysis of VE-cadherin in whole lung lysates from homozygous WT, KallR and K626/633R mice. α-tubulin is utilized as a loading control. VE-cadherin antibody (AF1002) against the N-terminal extracellular domain of the protein in (A, C). VE-cadherin antibody (C19) against the C-terminal domain of the protein in (B, D). (E, F) Quantification of VE-cadherin 100 kDa fragment as depicted in (A, C), presented as relative VE-cadherin 100 kDa fragment normalized to α-tubulin. $n = 7$ mice/group in (E) and $n = 10$ mice/group in (F). (G) HUVEC were depleted of endogenous VE-cadherin by siRNA transfection, followed by adenoviral expression of EGFP-tagged WT, K626/633R or KallR VE-cadherin constructs. Cells were treated with 100 nM bafilomycin or vehicle control for 6 h and lysed. Clarified lysates were analyzed by western blot with an anti-VE-cadherin antibody directed against the N-terminus of the protein (AF938), using α-tubulin as loading control. (H) Accumulation of N-terminal 110 kDa VE-cadherin fragments normalized to full-length protein levels and loading control. $n = 3$ biological replicates. Data information: In (E, F, H), data are presented as mean ± SEM. *$P < 0.05$; **$P < 0.01$; ****$P < 0.0001$; ns, not significant. Unpaired two-tailed $t$ test in (E, F, H). $P = 0.000000007$ (WT vs. KallR) in (E). $P = 0.00000001$ in (F). $P = 0.007482$ (WT vehicle vs. WT Bafilomycin), $P = K0.046290$ (K626/633 vehicle vs. K626/633 R Bafilomycin), $P = 0.034345$ (KallR vehicle vs. KallR Bafilomycin) and $P = 0.008421$ (WT Bafilomycin vs. K626/633R Bafilomycin) in (H). Source data are available online for this figure.

2023). In addition, LPS-induced degradation of VE-cadherin in endothelial cells was completely blocked and steady state levels of VE-cadherin were increased more than 3-fold in the lung vasculature of CHFR$^{\Delta EC}$ mice. Interestingly, we found no increase in the steady state expression level of VE-cadherin in the vasculature of cremaster muscle and lung even when all lysine residues of VE-cadherin were replaced by arginine. This may suggest that gene inactivation of CHFR may affect expression of VE-cadherin by yet additional mechanisms independent of the ubiquitination of VE-cadherin. In line with this, it is also

interesting that endothelial gene inactivation of CHFR blocked LPS-induced vascular permeability in the lung completely, whereas even the replacement of all lysine residues in the cytoplasmic domain of VE-cadherin had only a partial effect on histamine- and LPS- induced vascular permeability in skin and lung, respectively. This suggests that gene inactivation of CHFR affects endothelial junction integrity by additional targets, such as proteins that can associate with VE-cadherin (p120, β-catenin and VE-PTP), but also VE-cadherin independent proteins such as claudin 5 which were all strongly increased in lung lysates of CHFR$^{\Delta EC}$ (Tiruppathi et al,

2023). Other junctional antigens such as ESAM, JAM-A or nectins are potential candidates, but were not yet analyzed.

The fact that steady state expression levels of the K626/633R and KallR lysine mutants of VE-cadherin in our knock-in mice were indistinguishable from the expression of WT VE-cadherin in control mice, may suggest that ubiquitination of VE-cadherin is not relevant for steady state endocytosis and turnover of VE-cadherin. Alternatively, it is possible that reduced basal levels of steady state endocytosis of VE-cadherin may be compensated by feedback mechanisms that reduce biosynthesis levels or delivery to junctions.

Interestingly, immunoprecipitates of VE-cadherin from lung lysates of the different mouse lines contained higher levels of co-precipitated p120 for K626/633R and KallR VE-cadherin mutant mice, whereas co-precipitated levels of the different catenins which link cadherins to actin were similar for mutant and WT VE-cadherin. This is probably meaningful, since p120 has been implicated in regulating the cell surface levels of cadherins (Davis et al, 2003; Xiao et al, 2003; Chen et al, 2003). It was shown that p120 binds to a conserved membrane proximal site of cadherins which contain a motif of three acidic amino acids (DEE), which are needed for constitutive clathrin-dependent endocytosis of VE-cadherin (Nanes et al, 2012). Although this site is not involved in the K5 ubiquitin ligase-induced endocytosis of VE-cadherin, it was reported that p120 becomes displaced from VE-cadherin upon overexpression of K5 and K5-induced endocytosis of VE-cadherin was inhibited by p120 (Nanes et al, 2017). Thus, despite equal expression levels of WT and mutant VE-cadherin at junctions in vivo, the higher levels of associated p120 for the K/R mutants may suggest that steady state levels of ubiquitination of WT VE-cadherin may contribute to some level of p120 displacement and similarly to some level of constitutive endocytosis. Since we detected no difference in the expression levels of WT and K/R mutant VE-cadherin at junctions, we speculate that modulation of biosynthesis or cell surface transport efficiency of VE-cadherin may be compensatory.

In addition to the regulatory role of K626 and K633 ubiquitination in inflammation, we found that mutating these sites in vivo causes accumulation of intracellular VE-cadherin in venules, but not in arterioles. In line with this, we found that a 100 kDa VE-cadherin fragment accumulated in lung lysates of VE-cadherin K626/633R mice. This fragment was likely the equivalent of a 110 kDa N-terminal fragment of human VE-cadherin-K626/633R-EGFP, which accumulated in transduced HUVEC. A similar fragment accumulated for WT-VE-cadherin-EGFP in HUVEC only upon treatment with the lysosomal inhibitor bafilomycin. Collectively, these results suggest that ubiquitination of K626 and K633 may direct cleaved intracellular VE-cadherin to lysosomes during steady state turnover of VE-cadherin. Interestingly, the fact that intracellular accumulation of VE-cadherin K626/633R was only observed in venules and not in arterioles may indicate that junctional VE-cadherin has a higher steady state turnover rate in venules than in arterioles.

Intracellular accumulation of a similar large N-terminal fragment of VE-cadherin has been described before upon treatment of endothelial cells with chloroquine, another inhibitor of lysosomal activity (Su and Kowalczyk, 2017). This fragment was formed by calpain-mediated cleavage between the p120 and the β-catenin binding site. This cleavage occurred downstream of the

initiation of endocytosis and was blocked by mutating the DEE sequence motif needed for constitutive endocytosis of VE-cadherin (Su and Kowalczyk, 2017). Based on this, we assume that accumulation of this intracellular large N-terminal VE-cadherin fragment in venules of our VE-cadherin K/R knock-in mice could be due to constitutive DEE-dependent endocytosis which in combination with the lack of lysine residues leads to an accumulation of intracellular VE-cadherin since lysosomal targeting is inhibited. Our results imply that constitutive endocytosis is not impaired by mutating lysine residues in the cytoplasmic domain of VE-cadherin. Thus, ubiquitination of VE-cadherin may have a dual function, first triggering inflammation-induced endocytosis and second contributing to the targeting of VE-cadherin to lysosomes (Fig. EV3).

Together, our findings provide evidence that the two most membrane proximal lysine residues of VE-cadherin are required for ubiquitination, induced by various inflammatory mediators and for endocytosis and lysosomal targeting which are consequences of this modification. These two sites were in vivo relevant for histamine- and LPS-induced vascular permeability in skin and lung, respectively, as well as for the targeting of endocytosed intracellular VE-cadherin to lysosomes.

## Methods

### Cell culture and reagents

HUVEC were isolated from umbilical cords (Ethics Committee of Münster University Clinic Approval 2009-537-f-S) by treatment with 1 unit/ml Dispase II (Roche) for 10 min at 37 °C in M199 medium containing 1% penicillin/streptomycin, 20% fetal calf serum (FBS), 100 μg/ml heparin, and 3.1 μg/ml fungizone. HUVECs were cultured in EBM-2 medium supplemented with EGM-2 MV SingleQuots (Lonza) at 37 °C and 5% $CO_2$. Prior to experiments, HUVECs were starved as indicated in EBM-2 medium containing either 1% BSA or no BSA for different time periods and then treated with 100 nM bafilomycin A1 (Sigma-Aldrich), 100 μM histamine (Sigma-Aldrich) or 1 U/ml thrombin (Calbiochem).

The following point mutated human VE-cadherin adenoviral constructs, where lysine residues in the cytoplasmic tail of VE-cadherin were mutated to arginine, were created: K626R, K633R, K672R, K768R, K690R, the double mutants K626/633R and K712/713R and a mutant with all seven lysine residues replaced by arginine residues (KallR). A VE-cadherin EGFP cDNA (with the linker GCG CGA CCG GTC GCC ACC) connecting the carboxy-terminal Tyr 784 of VE-cadherin and the starting methionine of GFP, in the pENTR2B plasmid, was used to introduce the abovementioned mutations via the QuikChange Lightning Site-Directed Mutagenesis Kit (Agilent Technologies). Subsequently, the generated VE-cadherin K to R constructs were integrated into the pAD-DEST vector (Gateway Technology, Invitrogen) via LR recombinase reaction, to finally produce adenovirus in 293A cells (Invitrogen, R70507). The following additional reagents were used: Fluorescence mounting medium (Dako), 293A medium (DMEM, 10% FBS, 2 mM Glutamine, 1% Pen/Strep, 1% NEAA), Amersham ECL Western Blotting Detection Reagents (cytiva), SuperSignal West Femto Maximum Sensitivity Substrate (Thermo Scientific).

## Antibodies

The following commercial and previously generated antibodies were used (IF, immunofluorescence; WB, western blotting; IP, immunoprecipitation). Against mouse VE-cadherin: polyclonal antibody (pAb) C5 (prepared in-house (Gotsch et al, 1997)) (IP); pAb AF1002 (R&D Systems) (IF, WB), pAb C-19 (sc-6458; Santa Cruz) (WB); against human VE-cadherin: monoclonal antibody (mAb) Alexa Fluor 647 Mouse Anti-Human CD144 (clone 55-7H1; BD Pharmingen) (IF), mAb F-8 (sc-9989; Santa Cruz) (WB), pAb AF938 (R&D Systems) (WB); mAb to mouse PECAM-1 coupled to Alexa Fluor 555 (1G5.1 and 5D2.6; prepared in-house (Wegmann et al, 2006)) (IF); mAb to early endosome marker (EEA1, 3288, Cell Signaling) (IF); pAb to multivesicular body marker (IST-1, 19842-1-AP, Proteintech) (IF); pAb to GFP (ab6673; abcam) (IP); pAb to GFP (ab6556; abcam) (WB); pAb to p120-catenin S-19 (sc-1101, Santa Cruz); mAb to α-catenin (610194, BD Pharmingen); mAb to β-catenin (610154, BD Pharmingen); mAb to γ-catenin (610254, BD Pharmingen) and mAb to α-tubulin (T6074, Sigma-Aldrich); mAb to ubiquitin P4D1 (sc-8017; Santa Cruz) (all WB). Secondary antibodies were as follows: Alexa Fluor 488, 568 and 647-coupled secondary antibodies were purchased from Invitrogen (CA, USA) (IF). IRDye 680RD- and IRDye 800CW-coupled secondary antibodies were purchased from LI-COR Biosciences (WB). All other secondary antibodies were purchased from Jackson ImmunoResearch (WB).

## Generation of knock-in mice

The two knock-in mouse lines VE-cadherin KallR and K626/633R were designed by a recombinase mediated cassette exchange (RMCE) approach in which these constructs replace the wild-type VE-cadherin in mice. The QuikChange Lightning Site-Directed Mutagenesis Kit (Agilent Technologies) was used to insert the KallR and K626/633R mutations into the mouse VE-cadherin cDNA containing RMCE vector (pCR-Script URB3 exchange3 Tag U5-3), previously created in our laboratory (Schulte et al, 2011), containing a poly(A) transcriptional stop cassette and a hygromycin cassette flanked with flippase-recognition target sites. The complete insertion cassette contained a mutation specific tag for recognition by genotyping and was flanked by loxP and lox2272 sites. This allowed, with the help of Cre recombinase, to replace exon 2 of VE-cadherin in ES cells where this exon 2 was flanked by the same lox sites. In this way knock-in mice were generated principally similar as described before for other knock-in mutants of VE-cadherin created in our laboratory (Schulte et al, 2011; Broermann et al, 2011; Wessel et al, 2014). Genotyping was performed as described before (Schulte et al, 2011; Broermann et al, 2011; Wessel et al, 2014) and insertion of the desired mutation was verified by PCR-assisted sequencing of genomic DNA. To mitigate the possibility of potential side effects resulting from the expression of VE-cadherin through cDNA, control mice with a knocked-in VE-cadherin cDNA (WT KI) were used, previously generated and thoroughly characterized in our lab (Schulte et al, 2011). All mutated mice were homozygous for their mutation. All mice used in this study were either on mixed 129SVxC57Bl/6 genetic background: WT KI and KallR F1 generation (50% SV192 and C57Bl/6) and K626/633R F2 generation (25% SV129 and 75% C57Bl/6) or further backcrossed in the C57Bl/6 background: WT KI (99.9% C57Bl/6), KallR

and K626/633R F6 generation (98.44% C57Bl/6) or KallR F11 generation (99.95% C57Bl/6).

All mice were raised in the barrier facility of the Max Planck Institute for Molecular Biomedicine under pathogen-free conditions and given food and water ad libitum.

## Study approval

All animal experiments were approved by the Landesamt fuer Natur, Umwelt und Verbraucherschutz Nordrhein-Westfalen, Germany (Approval 81-02.04.2020.A187 and 81-02.04.2020.A369).

## Immunoprecipitation and immunoblot analysis

HUVECs were harvested in lysis buffer containing either 20 mM Tris-HCl, pH 7.4, 150 mM NaCl, 2 mM CaCl$_2$, 1 mM Na$_3$VO$_4$, 1% Triton X-100, 0.04% NaN$_3$ and 12.5 × cOmplete EDTA-free Protease Inhibitor (Roche) or 10 mM Na$_3$PO$_4$, 150 mM NaCl, 1% Nonidet P-40 substitute, 2 mM EDTA, and 12.5× cOmplete Protease Inhibitor (Roche), and lysed for 30 min at 4 °C followed by centrifugation for 30 min, 20,000× g at 4 °C. Aliquots were set aside for direct blot analysis and the remaining probe was used for immunoprecipitation with Protein G Sepharose 4 Fast Flow (GE Healthcare) and the respective antibody for 2 h at 4 °C. Extracted mice lungs were placed in RIPA lysis buffer (0,01 M NaPi (pH 7.2); 150 mM NaCl; 2 mM EDTA; 0,1% SDS; 1% NP-40; 1% Na-Deoxycholate; 1 mM DTT; 15× cOmplete Protease Inhibitor (Roche)), homogenized using the Precellys Evolution Touch Homogenizer with tissue homogenizing CKMix 2 ml tubes (Bertin Technologies), incubated for 3 h at 4 °C and centrifuged for 30 min at 20,000× g at 4 °C. Then aliquots were set aside for direct blot analysis. The remaining sample for immunoprecipitation (IP) was precleared for 1 h at 4 °C with Protein A Sepharose CL-4B (cytiva) and isotype antibody prior to IP over night at 4 °C with Protein A Sepharose and the respective antibody. Immunocomplex samples from HUVECs or mice lungs were washed five times with the respective lysis buffer and dissolved in sample buffer (200 mM Tris-HCl pH 6.8, 30% glycerol, 6% SDS, 0.1% bromophenol blue, and 150 mM dithiothreitol) for western blot analysis. Total HUVEC or whole lung lysates were loaded on 8% SDS-gels and immunoprecipitated probes on 6% SDS-gels for separation and further transfer to nitrocellulose by wet blotting. Blots were developed either with a film developer Curix 60 (Agfa) or with Odyssey CLx and Fc systems (Li-COR).

## RNA-mediated interference

Endogenous VE-cadherin expression was silenced in HUVEC by transfection with CDH5 siRNA (Life Technologies; 5′-GGGUU UUUGCAUAAUAAGCTT-3′). As a negative control, Allstars control siRNA (Qiagen) that does not target any known mammalian gene was used. HUVECs were transfected with 20-80 nM siRNA for 72 h using INTERFERin (Polyplus) or Lipofectamine RNAiMAX Transfection reagent (Invitrogen) according to manufacturer's instructions.

## Immunofluorescence staining

For analysis of VE-cadherin junction morphology in mouse venules and arterioles, whole mounts of mouse cremaster muscles,

diaphragm and cornea were stained. Homozygous WT KI, KallR or K626/633R mice were sacrificed by $CO_2$ inhalation, and perfused via the left heart ventricle with 1% paraformaldehyde (PFA) in PBS.

The cremaster muscle was dissected and left in situ for prefixation for 8 min with 4% PFA in PBS, before removal and fixation for another 1 h at room temperature (RT) in 4% PFA in PBS. Next, the cremaster muscle was permeabilized and blocked in 0.5% Triton X-100, 2% ovalbumine in PBS for 2 h at RT. Thereafter, the cremaster muscle was incubated with primary antibodies against the VE-cadherin extracellular domain, PECAM-1, early endosome (EE) or multivesicular body (MVB) markers followed by incubation with fluorescence labeled secondary antibodies.

The diaphragm was removed and prefixed for 10 min with 4% PFA in PBS, before 50 min incubation in cold 2% $H_2O_2$ in PBS, final fixation for 1 h at RT in 1% PFA, 0.1% Triton X-100, 0.1% NP-40 in PBS and blocked over night in 0.3% Triton X-100, 3% BSA in PBS at 4 °C. Thereafter, the diaphragm was incubated with primary antibodies against the VE-cadherin extracellular domain and PECAM-1 followed by incubation with fluorescence labeled secondary antibodies.

Whole mount cornea immunofluorescence staining and analysis were performed as previously described (Li et al, 2020).

Z-stack projections were acquired with a Zeiss LSM880 confocal microscope and analyzed with Fiji (Schindelin et al, 2012), Imaris (Oxford Instruments) and ZEN2010 software (Zeiss). Cremaster and Cornea whole mount staining for VE-cadherin and PECAM are depicted as maximum intensity projections. Diaphragm whole mount staining for VE-cadherin and PECAM as shown in Fig. 6A depict the "upper half" (closer to objective) of the vessel.

## Analysis of junctional VE-cadherin and VE-cadherin co-localization with EE or MVB

Quantification of junctional VE-cadherin in cremaster vessels was performed with an automated macro created with Fiji. PECAM-1 co-staining was used to create a mask of endothelial junctions from which the VE-cadherin fluorescence signal was quantified (to avoid detection of intracellular VE-cadherin) and normalized to vessel lumen.

To analyze VE-cadherin co-localization with EE or MVB markers in cremaster vessels, high resolution z-stack projections were acquired with a Zeiss LSM880 confocal microscope equipped with an airyscan module. Airyscan processing was performed with ZEN2010 software and co-localization analysis with Imaris. First, a vessel surface mask for the VE-cadherin, PECAM-1, EE or MVB channels was created. Second, PECAM-1 co-staining was used to create a mask of endothelial junctions and thereby excluding junctional VE-cadherin from the co-localization analysis. Third, an EE or MVB mask was created and finally, the VE-cadherin channel (excluding the junctional VE-cadherin staining) was overlayed with the EE or MVB mask and a threshold was set to determine co-localization of VE-cadherin with these compartments. Co-localization images shown in Fig. 7A,B depict the "upper half" (closer to objective) of the vessel.

## In vivo vascular permeability assay in the skin

A modified Miles assay for the induction of vascular permeability in the skin was performed as described previously (Wegmann et al, 2006). For each assay, three to five 10- to 12-week-old female mice were used. Evans blue dye (Sigma-Aldrich) was injected into the tail vein (100 µl of a 1% solution in PBS) and, 15 min later, 50 µl PBS or 175 ng histamine (Sigma-Aldrich) in 50 µl PBS was injected intradermally into the shaved back skin. After 30 min, skin areas were excised and extracted with formamide (Sigma-Aldrich) for 5 days, and the concentration of the dye was measured at 620 nm with a spectrophotometer (Shimadzu, UV-1900i).

## In vivo vascular permeability assay in the lung

LPS-induced vascular permeability in the mouse lung was assayed as previously described (Frye et al, 2015). Briefly, mice were exposed for 40 min to nebulized LPS (0.5 mg/ml Salmonella enteritidis, Sigma-Aldrich) or PBS as control. Five hours later, Evans blue dye (Sigma-Aldrich) was injected into the tail vein (20 mg/kg mouse) and after 30 min, mice were sacrificed and the lung circulation was perfused with PBS containing 2 mM EDTA (Roth). Lungs were removed and extracted with formamide (Sigma-Aldrich) for 5 days, and the concentration of the dye was measured at 620 nm with a spectrophotometer (Shimadzu, UV-1900i).

## In vitro permeability assay

To determine paracellular permeability, HUVECs, silenced for endogenous VE-cadherin expression, were seeded on 100 µg/mL fibronectin-coated Transwell filters (Costar, 6.5 mm diameter, 0.4 µm pore size; Corning). Forty-eight hours ahead of the experiment, VE-cadherin-EGFP WT, or K626/633R mutant constructs were expressed by adenoviral transduction. HUVECs were serum-starved in EBM-2 medium without supplements for 3 h, followed by stimulation with 1 U/ml thrombin parallel to the diffusion of 0.25 mg/ml FITC-dextran (250 kD; Sigma-Aldrich). After 30 min of incubation, a 50 µl aliquot of the medium containing the diffused fluorescent tracer was removed from the lower compartment and FITC fluorescence intensity was measured in a plate reader (Synergy 2, LabTek).

## Endocytosis assay

Endocytosis of VE-cadherin was assayed essentially as described (Nanes et al, 2012). HUVECs, silenced for endogenous VE-cadherin expression, were seeded on eight-well chamber slides (ibidi) coated with 100 µg/mL fibronectin (Sigma-Aldrich). Forty-eight hours ahead of the experiment, VE-cadherin-EGFP WT, Y685F or K626/633R mutant constructs were expressed by adenoviral transduction. HUVECs were serum-starved in EBM-2 medium without supplements for 3 h. Subsequently, VE-cadherin antibody (CD144, clone 55-7H1), which binds to the extracellular part of VE-cadherin and does not block cell adhesion, was added and cells were either stimulated with 100 µM histamine or left untreated for 1 h at 37 °C.

Afterwards, the remaining cell surface-bound antibodies were removed by acid wash (100 mM glycine-HCl, pH 2.5). Cells were fixed with 4% paraformaldehyde and permeabilized with 0.3% Triton X-100. Internalized VE-cadherin was labelled with an Alexa Fluor 568-coupled secondary antibody. Nuclei were stained with Hoechst 33342 (Invitrogen). Images were obtained using a Zeiss LSM880 confocal microscope. Endocytosed VE-cadherin

was quantified with an automated macro in Fiji (Xiao et al, 2003) as the number of internalized VE-cadherin vesicles per area normalized to the mean intensity of re-expressed VE-cadherin-EGFP. In total, 400–500 cells were evaluated per group per experiment.

## Analysis of VE-cadherin internalization by biotinylation

For evaluation of internalized VE-cadherin protein levels, HUVECs were depleted of endogenous VE-cadherin by siRNA transfection, followed by adenoviral re-expression of EGFP-tagged VE-cadherin WT, Y685F or K626/633R constructs 24 h later. In all, 100 mm dishes of subconfluent HUVECs were serum-starved in EBM-2 medium with 1% BSA for 16 h. Subsequently, cells were washed twice with ice-cold DPBS and treated with cell-impermeable biotinylation reagent (EZ-Link Sulfo-NHS-SS-Biotin, Thermo Fisher Scientific, 21331, 0.5 mg/ml in DPBS) for 30 min at 4 °C under gentle agitation. Cells were placed back on ice and unreacted biotinylation reagent was quenched by two-time washing with 100 mM glycine in DPBS. Cells were then treated with 100 μM histamine or vehicle for 60 min at 37 °C. Cells were transferred to ice, washed once with DPBS, and surface-bound biotin was cleaved by incubation with Bond-Breaker TCEP solution (Thermo Fisher Scientific, 77720, 25 mM in DPBS) for 30 min at 4 °C. Cells were lysed in 1 ml buffer containing 10 mM $Na_3PO_4$, 150 mM NaCl, 1% Nonidet P-40 substitute, 2 mM EDTA, and 12.5× cOmplete™ Protease Inhibitor (Roche) for 30 min at 4 °C. Clarified lysates were subjected to immunoprecipitation with neutravidin agarose beads (Thermo Fisher Scientific, 29202, 60 μL per sample).

## Experimental data and statistical analysis

Total sample numbers were determined on the basis of our previous studies with transgenic mouse models. Immunoblot signals were quantified using the software Image Studio™ (LI-COR Biosciences). For statistical analysis GraphPad Prism 9 software was used (GraphPad Software Inc.). Statistical significance was analyzed using the unpaired two-tailed t-test, one-way ANOVA followed by Holm–Sidak multiple comparison test or two-way ANOVA with Tukey's multiple comparison test. A *P* value of less than 0.05 was considered statistically significant. Data in graphs are presented as mean ± SEM.

## Data availability

This study includes no data deposited in external repositories.

The source data of this paper are collected in the following database record: biostudies:S-SCDT-10_1038-S44319-024-00221-7.

## Peer review information

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

## Acknowledgements

We thank Marika Meyer zu Brickwedde and Ute Ipe for excellent technical assistance. We are very grateful to the staff of the following core facilities of the Max Planck Institute for Molecular Biomedicine: the biooptics core facility for excellent help in image acquisition and evaluation, the transgenic core facility for great help in generating the genetically modified mouse lines and the animal care facility for thorough and reliable mouse breeding. This work was supported by a grant from the Deutsche Forschungsgemeinschaft (CRC1450, B03 to DV) and funds of the Max Planck-Gesellschaft (vascbiol A1 to DV).

## Author contributions

**Markus Wilkens**: Data curation; Formal analysis; Validation; Investigation; Writing–review and editing. **Leonie Holtermann**: Data curation; Formal analysis; Validation; Investigation; Writing–review and editing. **Ann-Kathrin Stahl**: Data curation; Formal analysis; Validation; Investigation. **Rebekka I Stegmeyer**: Data curation; Formal analysis; Validation; Investigation. **Astrid F Nottebaum**: Data curation; Formal analysis; Validation; Investigation; Project administration; Writing—review and editing. **Dietmar Vestweber**: Conceptualization; Supervision; Funding acquisition; Validation; Writing—original draft; Project administration; Writing—review and editing.

Source data underlying figure panels in this paper may have individual authorship assigned. Where available, figure panel/source data authorship is listed in the following database record: biostudies:S-SCDT-10_1038-S44319-024-00221-7.

## Funding

## Disclosure and competing interests statement

The authors declare no competing interests.

# Expanded View Figures

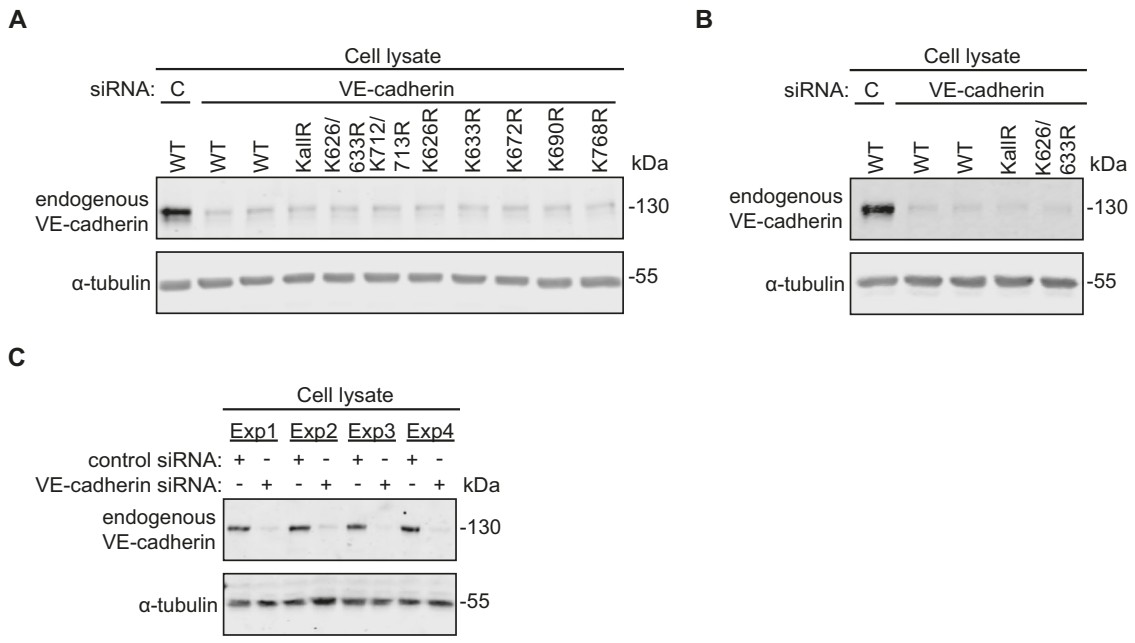

**Figure EV1. Endogenous VE-cadherin knockdown efficiency for experiments in Fig. 1.**

(A–C) Endogenous VE-cadherin was silenced in HUVEC via VE-cadherin siRNA. Whole cell lysates were immunoblotted for endogenous VE-cadherin and α-tubulin. Molecular sizes are indicated in kilodaltons (kDa). (A) Experiment shown in Fig. 1B. (B) Experiment shown in Fig. 1D. (C) Experiment shown in Fig. 1F.

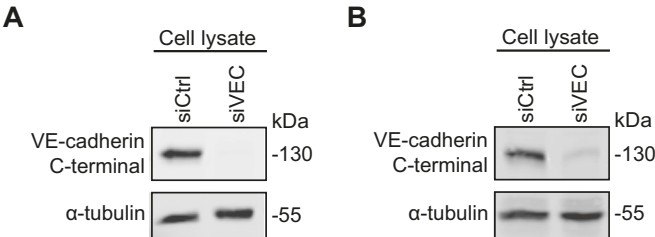

**Figure EV2. Endogenous VE-cadherin knockdown efficiency for experiments in Fig. 2.**

(**A, B**) Endogenous VE-cadherin was silenced in HUVEC via VE-cadherin siRNA. Whole cell lysates were immunoblotted for endogenous VE-cadherin and α-tubulin. Molecular sizes are indicated in kilodaltons (kDa). (**A**) Experiment shown in Fig. 2A. (**B**) Experiment shown in Fig. 2C.

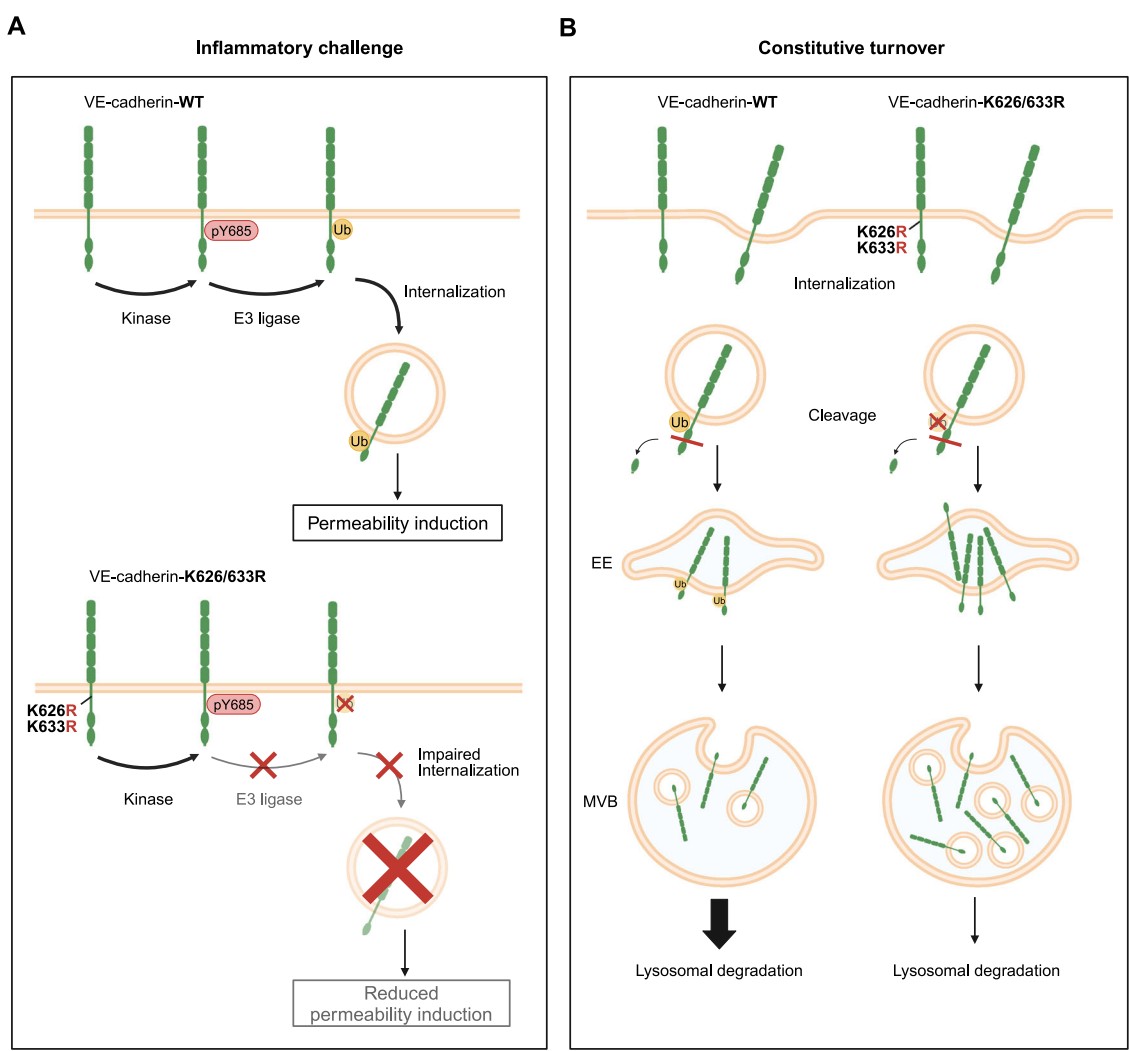

**Figure EV3.  Dual function of VE-cadherin ubiquitination in inflammation-induced endocytosis and lysosomal targeting.**

(A) Upon inflammatory challenge, Y685 of VE-cadherin gets phosphorylated followed by ubiquitination of VE-cadherin lysine residues K626 and K633, leading to VE-cadherin endocytosis and an increase in vascular permeability. This is prevented when K626 and K633 are mutated to arginine leading to impaired induction of vascular permeability. (B) Besides inflammation-induced endocytosis, ubiquitination of VE-cadherin controls a second process, the targeting of VE-cadherin to the lysosomal degradation pathway. Mutation of K626 and K633 to arginine slows down lysosomal targeting of mutated VE-cadherin leading to its accumulation in early endosomes (EE) and multivesicular bodies (MVB). Our results imply that K/R mutated VE-cadherin can be constitutively endocytosed in resting cells by lysine independent mechanisms, and therefore accumulates inside the cell, due to impaired degradation.

