## [Peer Review File · EMBO Reports]

Ubiquitination of VE-cadherin regulates inflammation-induced vascular permeability in vivo

Markus Wilkens, Leonie Holtermann, Ann-Kathrin Stahl, Rebekka Stegmeyer, Astrid Nottebaum, and Dietmar Vestweber

Corresponding author(s): Dietmar Vestweber (vestweb@mpi-muenster.mpg.de)

Review Timeline:

Submission Date:	22nd Nov 23
Editorial Decision:	13th Dec 23
Revision Received:	20th Jun 24
Editorial Decision:	16th Jul 24
Revision Received:	22nd Jul 24
Accepted:	23rd Jul 24

Editor: Achim Breiling

Transaction Report:

Dear Prof. Vestweber,

Thank you for the submission of your research manuscript to EMBO reports. I have now received the reports from the three referees that were asked to evaluate your study, which can be found at the end of this email.

As you will see, the referees think that the findings are of interest. However, they have several comments, concerns, and suggestions, indicating that a major revision of the manuscript is necessary to allow publication of the study in EMBO reports. As the reports are below, and all the referee concerns need to be addressed, I will not detail them here.

Given the constructive referee comments, I would like to invite you to revise your manuscript with the understanding that all referee concerns must be addressed in the revised manuscript or in a detailed point-by-point response. Acceptance of your manuscript will depend on a positive outcome of a second round of review. It is EMBO reports policy to allow a single round of revision only and acceptance of the manuscript will therefore depend on the completeness of your responses included in the next, final version of the manuscript.

- 1) a .docx formatted version of the final manuscript text (including legends for main figures, EV figures and tables), but without the figures included. Figure legends should be compiled at the end of the manuscript text.
- 2) individual production quality figure files as .eps, .tif, .jpg (one file per figure), of main figures (up to 8) and EV figures. Please upload these as separate, individual files upon re-submission.

- 4) a complete author checklist, which you can download from our author guidelines (<https://www.embopress.org/page/journal/14693178/authorguide>). Please insert page numbers in the checklist to indicate where the requested information can be found in the manuscript. The completed author checklist will also be part of the RPF.

- 5) that primary datasets produced in this study (e.g. RNA-seq, ChIP-seq, structural and array data) are deposited in an

appropriate public database. If no primary datasets have been deposited, please also state this in a dedicated section (e.g. 'No primary datasets have been generated and deposited'), see below.

The accession numbers and database should be listed in a formal "Data Availability" section (placed after Materials & Methods) that follows the model below. This is now mandatory (like the COI statement). Please note that the Data Availability Section is restricted to new primary data that are part of this study. This section is mandatory. As indicated above, if no primary datasets have been deposited, please state this in this section

Data availability

8) Regarding data quantification and statistics, please make sure that the number "n" for how many independent experiments were performed, their nature (biological versus technical replicates), the bars and error bars (e.g. SEM, SD) and the test used to calculate p-values is indicated in the respective figure legends (also for potential EV figures and all those in the final Appendix). Please also check that all the p-values are explained in the legend, and that these fit to those shown in the figure. Please provide statistical testing where applicable. Please avoid the phrase 'independent experiment', but clearly state if these were biological or technical replicates. Please also indicate (e.g. with n.s.) if testing was performed, but the differences are not significant. In case n=2, please show the data as separate datapoints without error bars and statistics. See also: <http://www.embopress.org/page/journal/14693178/authorguide#statisticalanalysis>

9) Please add scale bars of similar style and thickness to all the microscopic images, using clearly visible black or white bars (depending on the background). Please place these in the lower right corner of the images themselves. Please do not write on or near the bars in the image but define the size in the respective figure legend.

10) Please also note our reference format:

12) We now use CRedit to specify the contributions of each author in the journal submission system. CRedit replaces the author contribution section. Please use the free text box to provide more detailed descriptions and do not provide your final manuscript text file with an author contributions section. See also our guide to authors: <https://www.embopress.org/page/journal/14693178/authorguide#authorshipguidelines>

13) We would encourage you to use 'Structured Methods', our new Materials and Methods format. According to this format, the

Materials and Methods section should include a Reagents and Tools Table (listing key reagents, experimental models, software and relevant equipment and including their sources and relevant identifiers) followed by a Methods and Protocols section in which we encourage the authors to describe their methods using a step-by-step protocol format with bullet points, to facilitate the adoption of the methodologies across labs. More information on how to adhere to this format as well as downloadable templates (.doc or .xls) for the Reagents and Tools Table can be found in our author guidelines (section 'Structured Methods'):

14) Please order the manuscript sections like this, using these names:

Title page - Abstract - Keywords - Introduction - Results - Discussion - Materials and Methods - Data availability section - Acknowledgements - Disclosure and Competing Interests Statement - References - Figure legends - Expanded View Figure legends

I look forward to seeing a revised version of your manuscript when it is ready. Please let me know if you have questions or comments regarding the revision.

Yours sincerely,

Referee #1:

The manuscript by Wilkens et al from the Vestweber group examines the role of lysine residues in the VE-cadherin cytoplasmic tail, and the role of these proteins in vascular permeability. A range of approaches are taken, from site directed mutagenesis, cell culture models, endocytosis assays, and mouse models of vascular leak. This breadth of approaches is a strength of the study, as is the novel investigation of cadherin lysine residues/ubiquitination using in vivo models. In addition, the authors tie the K626 and K633 residues to previously described pathways involving Y685 phosphorylation, linking this phospho-regulation to ubiquitination. Overall, this is a very nice study and a well written manuscript.

The work could be improved with a few additional experiments to flesh out and bolster the authors' primary conclusions:

1) Figure 2 shows endocytosis assays and quantification of internalization using surface labeling of VE-cadherin. The increase in endocytosis is quite small, although it reaches statistical significance. In addition, the assay does not directly measure endocytosis because the surface pool of VE-cad antibody is not removed before quantification. Two approaches should be taken to strength this figure, which is essential to the proposed mechanism underlying the biology. First, acid wash should be done to remove VE-cad antibody from the cell surface so that puncta being counted are truly endocytic vesicles. Alternatives include EEA-1 colocalization or other means of surface stripping/quenching. Secondly, a biochemical approach such as biotinylation and cell surface stripping should be used to confirm the immunofluorescence assay.

2) Both thrombin and histamine are used to trigger VE-cadherin ubiquitination. In vitro barrier function assays using these mediators and the VE-cadherin mutants should be done to bolster the in vivo assays and interpretations.

3) In Figures 6 and 7, the authors should utilize markers for lysosomes and/or multivesicular bodies to determine if the VE-cadherin mutants lacking the lysine residues are being trapped in these compartments (or other endocytic compartments), as proposed.

Referee #2:

In this study, Wilkens et al provide valuable insights into the specific residues of lysine (K626 and K633) in VE-cadherin for its ubiquitination. Through a combination of in vitro and animal studies, the authors convincingly demonstrate the roles of K626 and K633 in the endocytosis and ubiquitination of VE-cadherin during histamine- or thrombin-induced inflammation. Overall, this study presents meaningful data, and the experimental execution is commendable.

However, the following key points should be considered:

1. The authors need to address or highlight why this study is important in understanding of molecular mechanism in vascular inflammation (permeability). Are there any patients who have mutants in these residues?

2. Although ubiquitin and ubiquitin-like modifications occur primarily on lysine residues of target proteins and stimulate a large number of downstream signals, the significance and purpose of this study are not clearly outlined in the Introduction or Discussion. It would be insightful if the authors included differential features and roles between VE-cadherin ubiquitination and other membrane-bound protein ubiquitination.
3. The authors claimed that the histamine-induced endocytosis of VE-cadherin is inhibited by Y685F and the K626/633R mutations (Fig. 2B, C). However, endocytosis of VE-cadherin, represented by 100 kD fragment, is increased by K626/633R and KallR mutants even without lysosomal inhibitor and in the lung lysates from K626/633R and KallR mice as well (Fig. 6B & 7B). Please provide an explanation for this apparent discrepancy.
4. How did the authors confirm the genetic changes of K626/633R and KallR mutant mice? Please clarify whether all mice used in this study were heterozygous or homozygous for the mutations.
5. The intracellular vesicles in Fig. 6A and 7A are presumably identical to the fragment of VE-cadherin (Fig. 6B and 7B), which are observed in the venule ECs of the cremaster. It would be valuable to clarify whether these intracellular vesicles are seen in the venules or sinusoids of other key organs such as the liver, lung, and bone marrow.
6. A graphical diagram for the novel findings of this study is required for the readers.

Referee #3:

The study by Wilkens et al. investigates a role of 7 lysine residues as potential ubiquitination sites leading to VE-cadherin internalization and lysosomal degradation upon endothelial cell stimulation with histamine or thrombin. The results demonstrate that the two most membrane proximal lysine residues of VE-cadherin are required for ubiquitination, induced by histamine and thrombin and for endocytosis and lysosomal targeting which are consequences of this modification. These two sites were in vivo relevant for histamine induced vascular permeability in the skin as well as for the targeting of endocytosed intracellular VE-cadherin to lysosomes. The study provides novel information about role of ubiquitination of VE-cadherin in control of vascular permeability in vitro and in vivo.

There are several points that need to be addressed to enhance the impact of this study.

1. What is the crosstalk between phosphorylation-dependent and ubiquitination-dependent VE-cadherin internalization and degradation, why both mechanisms are needed?
2. LPS induced lung permeability was completely blocked in CHFR EC mice, while histamine-induced EC permeability was only partially inhibited by mutation of ubiquitination sites on VE-cadherin. Can the authors discuss what other targets of CHFR might be involved in EC barrier disruption?
3. Histamine and thrombin cause GPCR-mediated acute and transient EC permeability. Clinically significant model is the one with more sustained inflammatory endothelial barrier dysfunction. LPS and other inflammatory pathogens trigger more complex signaling mechanisms activating both permeability and inflammation. What will be the effects of VE-cadherin mutants generated in this study on the magnitude of LPS- or bacteria-induced permeability and inflammation?
4. Some experiments in animal models of inflammatory edema, for example, LPS-induced pulmonary edema would significantly enhance potential clinical impact of these findings.
5. Bar graphs need to present individual points.
6. VE-cadherin endocytosis and permeability was blocked in both, Y685F and K626/633R mutants. Does this result demonstrate the superiority of Y685 phosphorylation over K626/633 ubiquitination as therapeutic target?

Point by point response to reviewer's comments

We thank the reviewers for their positive and very constructive comments that we have addressed below as follows:

Referee #1:

The manuscript by Wilkens et al from the Vestweber group examines the role of lysine residues in the VE-cadherin cytoplasmic tail, and the role of these proteins in vascular permeability. A range of approaches are taken, from site directed mutagenesis, cell culture models, endocytosis assays, and mouse models of vascular leak. This breadth of approaches is a strength of the study, as is the novel investigation of cadherin lysine residues/ubiquitination using in vivo models. In addition, the authors tie the K626 and K633 residues to previously described pathways involving Y685 phosphorylation, linking this phospho-regulation to ubiquitination. Overall, this is a very nice study and a well written manuscript.

The work could be improved with a few additional experiments to flesh out and bolster the authors' primary conclusions:

1) Figure 2 shows endocytosis assays and quantification of internalization using surface labeling of VE-cadherin. The increase in endocytosis is quite small, although it reaches statistical significance. In addition, the assay does not directly measure endocytosis because the surface pool of VE-cad antibody is not removed before quantification. Two approaches should be taken to strengthen this figure, which is essential to the proposed mechanism underlying the biology. First, acid wash should be done to remove VE-cad antibody from the cell surface so that puncta being counted are truly endocytic vesicles. Alternatives include EEA-1 colocalization or other means of surface stripping/quenching. Secondly, a biochemical approach such as biotinylation and cell surface stripping should be used to confirm the immunofluorescence assay.

As requested by the reviewer we have now performed the two additional endocytosis assays, using acid wash we removed anti VE-cadherin antibodies from the cell surface before quantification of endocytosed material (Fig. 2A + B). In addition, we used biotinylation and cell surface stripping by a reducing agent in order to determine endocytosis by a biochemical assay (Fig. 2C + D). Both these assays confirmed our original results.

2) Both thrombin and histamine are used to trigger VE-cadherin ubiquitination. In vitro barrier function assays using these mediators and the VE-cadherin mutants should be done to bolster the in vivo assays and interpretations.

As requested, we have now performed a permeability assay with 250 kD FITC dextran to determine whether the K/R mutations would affect thrombin-induced permeability across an endothelial cell monolayer. The results demonstrate that the K/R mutation affects

thrombin-induced monolayer permeability and are shown in figure 1F.

3) In Figures 6 and 7, the authors should utilize markers for lysosomes and/or multivesicular bodies to determine if the VE-cadherin mutants lacking the lysine residues are being trapped in these compartments (or other endocytic compartments), as proposed.

We have now double-stained endothelium in venules of the cremaster for VE-cadherin and either EEA1 or a marker for multivesicular bodies (Ist1). As shown in figures 7A and B, the VE-cadherin K/R mutants were more frequently found to localize to early endosomes or multivesicular bodies than WT VE-cadherin

Referee #2:

In this study, Wilkens et al provide valuable insights into the specific residues of lysine (K626 and K633) in VE-cadherin for its ubiquitination. Through a combination of *in vitro* and animal studies, the authors convincingly demonstrate the roles of K626 and K633 in the endocytosis and ubiquitination of VE-cadherin during histamine- or thrombin-induced inflammation. Overall, this study presents meaningful data, and the experimental execution is commendable.

However, the following key points should be considered:

1. The authors need to address or highlight why this study is important in understanding of molecular mechanism in vascular inflammation (permeability). Are there any patients who have mutants in these residues?

We have addressed this point in the first and second paragraph of the discussion. While it is widely believed that the regulation of VE-cadherin endocytosis is the major mechanism for the regulation of endothelial junctions, there is no direct evidence for this neither *in vitro* nor *in vivo*. Our study is important because it provides this direct evidence. Not only is it important to know that ubiquitination-dependent endocytosis of VE-cadherin is indeed significant and important *in vivo* for inflammation-induced vascular permeability. It is as important that completely blocking this mechanism (KallR mutation) does not completely block the induction of vascular permeability. Thus, endocytosis of VE-cadherin is only a part of the mechanism probably synergizing with other mechanisms, that might more directly regulate/modulate the adhesive strength of VE-cadherin (e.g. by actin linkage or clustering). Collectively, our study allows for the first time to judge the significance of VE-cadherin ubiquitination and endocytosis for the regulation of endothelial junction integrity *in vivo*.

As requested by the reviewer we searched in various data bases for potential patients with mutants in VE-cadherin lysine residues but did not find any. The following databases were searched: The Human Gene Mutation Database (<https://www.hgmd.cf.ac.uk/ac/index.php>), a cancer based database (GDC portal, <https://portal.gdc.cancer.gov/>) and the Uniprot variant viewer (<https://www.uniprot.org/uniprotkb/P33151/variant-viewer>).

2. Although ubiquitin and ubiquitin-like modifications occur primarily on lysine residues of target proteins and stimulate a large number of downstream signals, the significance and purpose of this study are not clearly outlined in the Introduction or Discussion. It would be insightful if the authors included differential features and roles between VE-cadherin ubiquitination and other membrane-bound protein ubiquitination.

We have now inserted a paragraph on the last page of the introduction that refers to the various mechanisms in cells that are regulated by ubiquitination and mention that VE-cadherin endocytosis can be regulated by ubiquitin dependent and independent mechanisms.

3. The authors claimed that the histamine-induced endocytosis of VE-cadherin is inhibited by Y685F and the K626/633R mutations (Fig. 2B, C). However, endocytosis of VE-cadherin, represented by 100 kD fragment, is increased by K626/633R and KallR mutants even without lysosomal inhibitor and in the lung lysates from K626/633R and KallR mice as well (Fig. 6B & 7B). Please provide an explanation for this apparent discrepancy.

Our results suggest that the K626/633R mutation has a dual effect, one on endocytosis of VE-cadherin from the cell surface into endocytic vesicles. The second is linked to a later step in intracellular translocation of VE-cadherin: the targeting to lysosomes. This is the reason why we detect accumulation of the 100 kD fragment of VE-cadherin and detect increased staining for intracellular VE-cadherin at early endosomes and multivesicular bodies, as we now show in the new figure 7. It is known that constitutive endocytosis of VE-cadherin depends on motifs (DEE) within the cytosolic domain of VE-cadherin, which are independent of lysine residues (reported in the discussion part on pages 13 and 14). Our results imply that this constitutive endocytosis occurs steadily even when the lysine residues are mutated. Since mutating the lysine residues of VE-cadherin impairs lysosomal targeting of VE-cadherin, constitutively endocytosed VE-cadherin accumulates in the cells. Thus, accumulation is not due to more endocytosis of the lysine mutants, but due to impaired degradation of constitutively endocytosed VE-cadherin. Our results imply that constitutive endocytosis is not impaired by mutating lysine residues in the cytoplasmic domain of VE-cadherin, which is now stated explicitly in the discussion on page 14, the legend of Fig. EV3 (graphical abstract) and was already mentioned in the last sentence of the results section.

4. How did the authors confirm the genetic changes of K626/633R and KallR mutant mice? Please clarify whether all mice used in this study were heterozygous or homozygous for the mutations.

To confirm that the K626/633R and KallR mutant mice contain the correct K/R mutations, genomic DNA of homozygous mice was isolated and insertion of the desired mutation was verified by PCR-assisted sequencing of genomic DNA. This is now stated in M&M. All mice used in this study were homozygous for the respective mutations which was verified by genotyping for the RMCE insertion cassette as was done for other VE-cadherin knock in mutants before (references 7, 8 and 24) and as is stated in M&M.

5. The intracellular vesicles in Fig. 6A and 7A are presumably identical to the fragment of VE-cadherin (Fig. 6B and 7B), which are observed in the venule ECs of the cremaster. It would be valuable to clarify whether these intracellular vesicles are seen in the venules or sinusoids of other key organs such as the liver, lung, and bone marrow.

Indeed, we think that a significant fraction of the intracellular, endocytosed VE-cadherin exists as a large 100 kD fragment. As requested, we have now analyzed whether these vesicles are also seen in venules of other organs. As shown in Fig. 6A and B, we could identify such cytosolic vesicles in diaphragm and cornea.

6. A graphical diagram for the novel findings of this study is required for the readers.

A graphical diagram was prepared and is now Fig. EV3 of the manuscript.

Referee #3:

The study by Wilkens et al. investigates a role of 7 lysine residues as potential ubiquitination sites leading to VE-cadherin internalization and lysosomal degradation upon endothelial cell stimulation with histamine or thrombin. The results demonstrate that the two most membrane proximal lysine residues of VE-cadherin are required for ubiquitination, induced by histamine and thrombin and for endocytosis and lysosomal targeting which are consequences of this modification. These two sites were in vivo relevant for histamine induced vascular permeability in the skin as well as for the targeting of endocytosed intracellular VE-cadherin to lysosomes. The study provides novel information about role of ubiquitination of VE-cadherin in control of vascular permeability in vitro and in vivo.

There are several points that need to be addressed to enhance the impact of this study.

1. What is the crosstalk between phosphorylation-dependent and ubiquitination-dependent VE-cadherin internalization and degradation, why both mechanisms are needed?

We assume that tyrosine phosphorylation of VE-cadherin acts as an upstream signal to trigger ubiquitination of VE-cadherin. Indeed, Orsenigo et al showed (reference 23) that bradykinin-induced ubiquitination of VE-cadherin was impaired for the Y685F VE-cadherin mutant. This was mentioned in the introduction (page 4, first paragraph). It was originally shown for growth factor receptors that their phosphorylation at tyrosine residues is linked to and followed by ubiquitination. This way receptor activation is linked to ubiquitination triggered endocytosis or intracellular targeting (Levkowitz et al., Mol. Cell, 1999, 4: 1029-1040)

2. LPS induced lung permeability was completely blocked in CHFR Δ EC mice, while histamine-induced EC permeability was only partially inhibited by mutation of ubiquitination sites on

VE-cadherin. Can the authors discuss what other targets of CHFR might be involved in EC barrier disruption?

We have discussed the fact that gene inactivation of CHFR completely blocks LPS-induced induction of vascular permeability, whereas even the mutation of all lysine residues only has a partial inhibitory effect. This clearly suggests that CHFR deletion increases endothelial junction integrity by yet other targets such as proteins that can associate with VE-cadherin (p120, β -catenin and VE-PTP), but also VE-cadherin independent proteins such as claudin 5 which were all strongly increased in lung lysates of CHFR ^{Δ EC} [30]. Other junctional antigens such as ESAM, JAM-A or nectins are potential candidates, but were not yet analyzed. We have added these comments to the discussion (pages 12/13).

3. Histamine and thrombin cause GPCR-mediated acute and transient EC permeability. Clinically significant model is the one with more sustained inflammatory endothelial barrier dysfunction. LPS and other inflammatory pathogens trigger more complex signaling mechanisms activating both permeability and inflammation. What will be the effects of VE-cadherin mutants generated in this study on the magnitude of LPS- or bacteria-induced permeability and inflammation?

As this reviewer suggested in the next point (point 4), we have now tested whether the magnitude of LPS-induced vascular permeability in the lung would be reduced in K626/633R and KallR mice and found strong, although again no complete inhibition of leak induction (Figures 4 E and F). Since it took us many more mice than anticipated to establish this assay, we ran into breeding problems and limitations of the number of mice approved for such experiments. Therefore, we refrained from performing additional experiments to determine inflammatory parameters other than permeability.

4. Some experiments in animal models of inflammatory edema, for example, LPS-induced pulmonary edema would significantly enhance potential clinical impact of these findings.

As described under point 3, these experiments have now been performed (Fig. 4E and F).

5. Bar graphs need to present individual points.

All bar graphs have been changed accordingly.

6. VE-cadherin endocytosis and permeability was blocked in both, Y685F and K626/633R mutants. Does this result demonstrate the superiority of Y685 phosphorylation over K626/633 ubiquitination as therapeutic target?

The Y685F as well as the K626/633R mutation each reduced endocytosis of VE-cadherin and vascular permeability induction to a similar extent. This had been expected since Y685 phosphorylation acts upstream of lysine ubiquitination. Whether any of the two modifications is better suited than the other as therapeutic target is an open question.

To interfere with VE-cadherin ubiquitination would require to target the relevant E3 ligase and to test side-effects via other substrates. To target Y685 phosphorylation would require

to target the relevant kinase. Since it is likely that this is one of the members of the Src kinase family, it is questionable that such an approach would be sufficiently specific.

A much more specific approach would be to stabilize the interaction between VE-cadherin and the associated tyrosine phosphatase VE-PTP which targets Y685 of VE-cadherin. We have previously successfully pursued such an approach. We generated knock in mice that expressed the fusion proteins VE-cadherin-FK 506 binding protein and VE-PTP-FRB* under the control of the endogenous VE-cadherin promoter, thus replacing endogenous VE-cadherin. The additional domains in both fusion proteins allow the heterodimeric complex to be stabilized by a chemical compound (rapalog). We found that intravenous application of the rapalog strongly inhibited VEGF-induced (skin) and LPS-induced (lung) vascular permeability (Broermann et al., *J Exp Med*, 208:2393-2401, 2011).

To achieve stabilization of the VE-cadherin/VE-PTP complex in WT mice, one should generate bi-specific antibodies that stabilize the interaction of both proteins. Personally, I think such an approach would be feasible and would be a very specific way to target and prevent phosphorylation of Y685 of VE-cadherin. In addition, we assume that VE-PTP may support the function of VE-cadherin by additional mechanisms, which we are presently analyzing.

Dear Prof. Vestweber,

Thank you for the submission of your revised manuscript to our editorial offices. I have now received the reports from the three referees that I asked to re-evaluate the study, you will find below. As you will see, the referees now fully support the publication of the study in EMBO reports. Referee #1 has a remaining concern I ask you to address in a final revised manuscript. Please also provide a final p-b-p-response regarding this point of the referee.

- I would suggest this slightly modified title:

Ubiquitination of VE-cadherin regulates inflammation-induced vascular permeability in vivo.

- Please provide the abstract written in present tense throughout.

- Please order the manuscript sections like this, using these names:

Abstract - Keywords - Introduction - Results - Discussion - Methods - Data availability section - Acknowledgements - Disclosure and Competing Interests Statement - References - Figure legends - Tables - Expanded View Figure legends

- Please note that the legends for figures 3f-i is not provided in a sequential manner (legend for figure 3h, is provided before the legends of figure 3f, g). This needs to be rectified.

- Please note that the exact p values are not provided in the legends of figures 1c, e-f; 2b; 4a-f; 8e-f. Please do that.

- Please make sure that all the funding information is also entered into the online submission system and that it is complete and similar to the one in the acknowledgement section of the manuscript text file. Please note that the Max Planck-Gesellschaft needs to be removed from the Comments box in the submission system and needs to be entered as a separate Funder.

- Please make sure that all figure panels and tables are called out separately and sequentially. Presently, panels 4A-D are called out before panel 3A; panels 8A and 8C are called out before those of Fig. 7. Moreover, there seems to be no callout for panel 8B. Please check.

- I would suggest, as they are small, to turn the Supplemental Tables into main tables. Otherwise, they would need to be moved into an Appendix file. To have them as main tables, please rename them to Table 1-3 and update their callouts in the manuscript text. The heading 'Supplemental Tables' should then be changed to 'Tables'.

- Please use our reference format:

In addition, I would need from you:

- a schematic summary figure as separate file that provides a sketch of the major findings (not a data image) in jpeg or tiff format (with the exact width of 550 pixels and a height of not more than 400 pixels) that can be used as a visual synopsis on our website.

Best,

Referee #1:

The manuscript by Wilkens and colleagues from the Vestweber laboratory has been revised and substantially improved from the original submission. My only remaining concern is the interpretation of the experiments in Fig. 2A,B where the authors measure endocytosis of WT and mutant VE-cadherin in response to histamine. The inclusion of surface labeling (fluorescence and biotinylation) is an important addition. However, the 60 min time course for internalization is complicating the experiment because some of the cadherin is already entering degradative compartments by this time. Evidence for this possibility is the

increase in endocytic vesicles observed in the K626/633R mutant under control conditions, compared to WT. A shorter time course, in the range of 15-30 min, would have been preferable for both the fluorescence based and biotinylation based approaches. But perhaps not enough internalization would be observed at earlier time points to yield reliable quantification? At any rate, the authors mention that trafficking to degradative compartments likely explains the results, and most readers will appreciate this important point of clarification.

Referee #2:

The authors adequately addressed the comments. It is now ready to be published.

Referee #3:

The authors adequately addressed the concerns raised in the previous review and performed additional studies suggested by this referee. The manuscript has been substantially improved and is now suitable for publication in EMBO Reports.

Point by point response to reviewer's comments

We thank the reviewers for their positive and very constructive comments that we have addressed below as follows:

Referee #1:

The manuscript by Wilkens and colleagues from the Vestweber laboratory has been revised and substantially improved from the original submission. My only remaining concern is the interpretation of the experiments in Fig. 2A,B where the authors measure endocytosis of WT and mutant VE-cadherin in response to histamine. The inclusion of surface labeling (fluorescence and biotinylation) is an important addition. However, the 60 min time course for internalization is complicating the experiment because some of the cadherin is already entering degradative compartments by this time. Evidence for this possibility is the increase in endocytic vesicles observed in the K626/633R mutant under control conditions, compared to WT. A shorter time course, in the range of 15-30 min, would have been preferable for both the fluorescence based and biotinylation based approaches. But perhaps not enough internalization would be observed at earlier time points to yield reliable quantification? At any rate, the authors mention that trafficking to degradative compartments likely explains the results, and most readers will appreciate this important point of clarification.

We agree that the amount of internalized VE-cadherin is always a net result of uptake and degradation. Since the K/R mutation slows down lysosomal targeting and degradation, we see that VE-cadherin-K626/633R accumulates in comparison to the WT form of VE-cadherin. This is observed under control conditions (no histamine), as the reviewer pointed out and as we had stated on page 7 of our recent revised manuscript (last two sentences of endocytosis chapter). In the presence of histamine, we see an increase of internalized intracellular WT VE-cadherin (due to endocytosis), but no further increase of intracellular K626/633R VE-cadherin. Since histamine triggers endocytosis but not lysosomal targeting, we believe that our results show that the K626/633R mutation inhibits endocytosis. Otherwise, a further increase of internalized molecules would have been detected. We agree that a shorter time point (like 30 min) would have resulted in less accumulation due to reduced lysosomal targeting. However, this early time point would have also resulted in less detection of endocytosed material. Since it was important to analyze conditions at which we see a highly significant increase of intracellular WT VE-cadherin with histamine compared to control, we choose the 60 min time point. We appreciate that the reviewer welcomed our mentioning of the fact that trafficking "to degradative compartments likely explains the results, and most readers will appreciate this important point of clarification".

Referee #2:

The authors adequately addressed the comments. It is now ready to be published.

We thank the referee for this recommendation.

Referee #3:

The authors adequately addressed the concerns raised in the previous review and performed additional studies suggested by this referee. The manuscript has been substantially improved and is now suitable for publication in EMBO Reports.

We thank the referee for this recommendation.

Prof. Dietmar Vestweber
Max-Planck-Institute for molecular Biomedicine
Dpt. of Vascular Cell Biology
Röntgenstr. 20
MÃ1/4nster D-48149
Germany

Dear Prof. Vestweber,

I am very pleased to accept your manuscript for publication in the next available issue of EMBO reports. Thank you for your contribution to our journal.

Yours sincerely,
